



# Continuous secondary ice production initiated by updrafts through the melting layer in mountainous regions

Annika Lauber[1], Jan Henneberger[1], Claudia Mignani[2], Fabiola Ramelli[1], Julie T. Pasquier[1], Jörg Wieder[1], Maxime Hervo[3], and Ulrike Lohmann[1]

[1]ETH Zurich, Institute for Atmospheric and Climate Science, Zurich, Switzerland
[2]Department of Environmental Sciences, University of Basel, Basel, Switzerland
[3]Federal Office of Meteorology and Climatology MeteoSwiss, Payerne, Switzerland

**Correspondence:** Annika Lauber (annika.lauber@env.ethz.ch), Jan Henneberger (jan.henneberger@env.ethz.ch)

**Abstract.** An accurate prediction of the ice crystal number concentration in clouds is important to determine the radiation budget, the lifetime, and the precipitation formation of clouds. Secondary ice production is thought to be responsible for the observed discrepancies between the ice crystal number concentration and the ice nucleating particle concentration in clouds. The Hallett-Mossop process is active between -3 °C and -8 °C and has been implemented into several models while all other

secondary ice processes are poorly constrained and lack a well-founded quantification. During two hours of measurements taken on a mountain slope just above the melting layer at temperatures warmer than -3 °C, a continuously high concentration of small plates identified as secondary ice was observed. The presence of drizzle drops suggests droplet fragmentation upon freezing as the responsible secondary ice mechanism. The constant supply of drizzle drops can be explained by a recirculation theory, suggesting that melted snowflakes, which sedimented through the melting layer, were reintroduced into the cloud

as drizzle drops by orographically forced updrafts. Here we introduce a parametrization of droplet fragmentation at high temperatures when primary ice nucleation is basically absent and the first ice is initiated by collision of drizzle drops with aged ice crystals sedimenting from higher altitudes. Based on previous measurements, we estimate that a droplet of 200 μm in diameter produces 18 secondary ice crystals when it fragments upon freezing. The application of the parametrization to our measurements shows high uncertainties, but the estimated number of splinters produced per fragmenting droplet (18-43) lies

within the range of uncertainty if we assume that all droplets larger than 40 μm fragment when they freeze.

## 1 Introduction

Accurate weather forecasting in mountainous regions is more challenging than over flat topography as orography has a strong influence on the local weather, e.g. it creates local up- and downdrafts, which strongly impacts the development of clouds (Roe, 2005; Henneberg et al., 2017). Yet, orography is not very well resolved in numerical weather prediction models because

the high resolution needed requires a lot of computational power that is often not available (e.g., Fundel et al., 2010; Gowan et al., 2018). Furthermore, microphysical processes in clouds, which affect the development of the concentration and size of ice crystals and cloud droplets, are still not completely understood (e.g., Field et al., 2017). However, the correct phase partitioning and concentration of cloud particles is important for the determination of the radiation budget, the lifetime and





the precipitation amount of clouds (e.g., Lohmann, 2002; Henneberg et al., 2017). This can be especially important for mixed-phase clouds (MPCs) consisting of ice crystals and liquid droplets since they have a major contribution to the total precipitation in the midlatitudes (Mülmenstädt et al., 2015).

MPCs are thermodynamically unstable because the saturation water vapor pressure over ice is lower than over liquid water, allowing ice crystals to grow faster than liquid droplets. The particle growth reduces the water vapor pressure and eventually leads to the evaporation of cloud droplets when the water vapor pressure drops below water vapor saturation over liquid. This process is called the Wegener-Bergeron-Findeisen process (Wegener, 1911; Bergeron, 1935; Findeisen, 1938) and can lead to rapid glaciation of clouds. Nevertheless, persistent MPCs have been frequently observed in mountainous regions, where the local topography produces sufficiently large updrafts to sustain a continuous condensate production (e.g., Korolev, 2007; Lohmann et al., 2016). An updraft velocity of about $2\,\mathrm{m\,s^{-1}}$ is often sufficient to maintain supersaturation with respect to water, which enables a simultaneous growth of cloud droplets and ice crystals (Korolev, 2007).

MPCs exist between 0 °C and -38 °C. In this temperature range, ice nucleating particles (INPs) are required for cloud droplets to freeze. The resulting so-called primary ice can create additional ice crystals by any kind of fragmentation referred to as secondary ice production (SIP) (e.g., Field et al., 2017). Assuming only primary ice to exist, one would expect that the ice crystal number concentration (ICNC) equals the activated ice nucleating particle concentration (INPC) in clouds. This is for example observed in shallow stratocumulus (Mossop et al., 1972) or thin stratus clouds but often does not hold true in more convective clouds. A discrepancy between the INPC and the ICNC of up to 4 orders of magnitude has been observed in several studies (e.g., Koenig, 1963; Hobbs and Rangno, 1985; Crawford et al., 2012; Lasher-Trapp et al., 2016; Ladino et al., 2017; Beck et al., 2018). If surface-based processes like blowing snow (e.g., Beck et al., 2018) and the seeder-feeder process (hydrometeors which formed aloft precipitate into the cloud below; e.g., Bader and Roach (1977); Ramelli et al. (2020)) can be excluded as a potential ice source, this discrepancy can only be explained by SIP.

On 22 February 2019, during the RACLETS (Role of Aerosols and CLouds Enhanced by Topography on Snow) campaign in the region of Davos in the Swiss Alps (also introduced in Ramelli et al. (2020)), a discrepancy between the INPC and ICNC of several orders of magnitude was measured. In the following subsection different SIP mechanisms, which might be responsible for this high discrepancy, will be described. Furthermore, a recirculation theory by Korolev et al. (2020) will be introduced that can explain the observations.

## 1.1 Secondary ice production mechanisms

Six SIP mechanisms have been discussed over the past decades. They are expected to be active depending on the environmental conditions like temperature, cloud particle size distribution and updrafts: (i) the rime-splintering or Hallett-Mossop process, (ii) collisional breakup, (iii) droplet fragmentation upon freezing, (iv) thermal shock fragmentation, (v) fragmentation of sublimating ice and (vi) activation of INPs in transient supersaturation of freezing droplets (Korolev and Leisner, 2020).

The rime-splintering process describes the production of secondary ice during riming. Yet, the underlying physical mechanism of this process is not well defined (Field et al., 2017). The most common explanation for SIP during riming is that droplets, which freeze on an ice crystal, build up an internal pressure and break up upon freezing. In a laboratory study, Hallett




and Mossop (1974) observed that rime-splintering is active at temperatures between -3 °C and -8 °C when particles grow by riming. Dong and Hallett (1989) showed that at temperatures above -3 °C, droplets spread out on the ice surface and do not build an ice shell, while at temperatures below -8 °C the ice shell of rime might be too strong to break (Griggs and Choularton, 1983). While several observations of secondary ice could be explained by the rime-splintering process (e.g., Ono, 1971, 1972;

Harris-Hobbs and Cooper, 1987; Bower et al., 1996), secondary ice was also observed in cases where these requirements were not fulfilled (e.g., Korolev et al., 2020) or the SIP by rime-splintering alone was expected to be too slow to explain the observed rapid glaciation (e.g., Hobbs and Rangno, 1990).

Collisional breakup describes the break-up of an ice crystal due to a collision with another ice crystal. This process was shown to be active when large ice crystals (e.g., graupel or aggregates) are present (Korolev and Leisner, 2020). Fragments of

ice crystals were observed in several field studies (e.g., Jiusto and Weickmann, 1973; Schwarzenboeck et al., 2009). Vardiman (1978) found that significant SIP occurs by collisional breakup, when relatively large concentrations of rimed ice crystals are present in convective clouds. He concluded that collisional breakup cannot explain high secondary ICNC at warm temperatures. The laboratory study of Takahashi et al. (1995) showed that a maximum SIP rate occurs at -16 °C when ice spheres with different growth modes collide.

Droplet fragmentation during freezing can happen when liquid water gets trapped inside a freezing droplet after an ice shell formed around the droplet and expands. Due to the lower density of ice than liquid water, an internal pressure builds up. If the pressure reaches a critical point, different processes have been observed to release the pressure: a complete breakup in mainly two halves, opening and closing of cracks, bubbles forming and bursting on the ice shell and streams being ejected from the shell, which may contain small ice fragments (Lauber et al., 2018). An average number of splinters being ejected per

freezing droplet could not be quantified so far because many ejected splinters are expected to be too small and the processes happen so fast that the used measurement techniques until today were not able to detect all splinters (Lauber et al., 2018; Keinert et al., 2020). Keinert et al. (2020) showed that droplets freezing in moving air break up one order of magnitude more frequently than droplets freezing in stagnant air. Therefore, high wind speed and turbulence can be expected to increase the likelihood of droplet fragmentation. The highest fragmentation rate was observed around -15 °C. It decreases to higher and

lower temperatures, while it seems to increase again at temperatures close to 0 °C (Keinert et al., 2020). The larger the droplet, the more likely it will fragment and the more splinters it will likely produce (Kolomeychuk et al., 1975; Lauber et al., 2018). Many field studies showed that large droplets are present in clouds before ICNCs exceed the INPCs by orders of magnitude, (e.g., Koenig, 1963; Braham, 1964; Mossop et al., 1970; Hobbs and Rangno, 1990; Rangno, 2008; Lawson et al., 2017; Korolev et al., 2020) and often explained these observations with SIP by droplet fragmentation. Even though large droplets are

rare in clouds, they might start a cascading process of splinter production when produced splinters hit other droplets, which subsequently freeze and produce more splinters (Koenig, 1963; Chisnell and Latham, 1974; Lawson et al., 2015).

Thermal shock fragmentation (King and Fletcher, 1976) and the activation of INPs in transient supersaturation of freezing droplets both require precipitation size droplets to be active, while the fragmentation of sublimating ice can be important if cloud regions are affected by entrainment of dry air (Korolev and Leisner, 2020).





From the six SIP mechanisms, the rime-splintering process is the best constrained mechanism and has been implemented in some cloud microphysics schemes (e.g., Scott and Hobbs, 1977; Beheng, 1987; Phillips et al., 2001). There have been fewer attempts to include collisional breakup (e.g., Yano and Phillips, 2011) and droplet fragmentation (e.g., Lawson et al., 2015). Sullivan et al. (2018) modeled all three ice multiplication processes and showed that none of the three SIP mechanisms

dominates the ICNC enhancement. While collisional breakup is important when low updrafts prevail and high nucleation rates are present, droplet fragmentation and the rime-splintering process can also be important in INP limited regions. However, they pointed out that the processes are not very well constrained by laboratory and in situ data.

## 1.2 Persistent SIP immediately above the melting layer

A remarkable finding of the study by Korolev et al. (2020) is the observation of small pristine ice crystals persisting immediately

above the melting layer in clouds. They calculated a spatial correlation time $\tau_{\mathrm{corr}}$ during which the environmental changes (e.g., air temperature, humidity and cloud particle number concentration) are insignificant and the shapes of the ice crystals can still be associated with the environment they are growing in. Based on their estimation, $\tau_{\mathrm{corr}}$ is on the order of 60-120 s. Assuming water vapor saturation over liquid, an ice column at temperatures below -3 °C can reach a length between 50 and 150 µm during $\tau_{\mathrm{corr}}$ depending on its aspect ratio. A relative humidity close to water saturation is a valid assumption in MPCs (Korolev and

Isaac, 2006). It can be concluded that if external processes like blowing snow or the seeder-feeder process can be excluded, the discrepancy between the INPC and the concentration of columns shorter than 50 to 150 µm in MPCs must emerge from SIP.

Korolev et al. (2020) suggested a recirculation process through the melting layer to be a possible explanation for the observed high concentration of small pristine ice crystals. Ice crystals, which fall through the melting layer as precipitation, melt into drizzle sized droplets. If convection or turbulence causes large updrafts, these droplets can be reintroduced into the cloud.

Whenever these droplets freeze by the collision with ice crystals, the droplet fragmentation upon freezing process can become active and produce high amounts of secondary ice particles.

## 2 Experimental setup

### 2.1 RACLETS campaign

The measurements used in this study were collected during the RACLETS campaign, which took place in February and March

2019 in the region of Davos in the Swiss Alps. The objective of this campaign was to improve the understanding of the influence of topography and aerosols on the development of clouds. For this goal, a set of instruments was deployed at different locations. Fig. 1 shows the locations and instruments, which were used for the analysis of the presented case study.

Measurements of the in-cloud properties such as the cloud particle concentration as well as their size distribution and shape were taken from the HoloGondel platform (Beck et al., 2017) on the Gotschnabahn, which is described in more detail in

section 2.2. As part of the MeteoSwiss observation network, a ceilometer (Vaisala, Model CL31) was installed in Klosters (1200 m) to determine the cloud base height (Hervo, 2020a) as well as a wind profiler (Vaisala, Model Lap3000, Finland) in





Wolfgang to measure the horizontal wind field (Hervo, 2020b). The general wind pattern on the ground was determined by data from different MeteoSwiss stations, the Snow and Avalanche Research SLF and the Holfuy station of the paragliding club Grischna. Temperature measurements were taken from the MeteoSwiss station in Klosters and the snow drift station installed at Gotschnagrat (Walter et al., 2020). For the analysis of the whole clouds, a vertically-pointing cloud radar (Model

Mira-36, METEK GmbH, Germany; Görsdorf et al. (2015)) was installed in Wolfgang. INPCs were measured in the valley in Wolfgang (1630 m) and on the mountain top of Weissfluhjoch (2670 m), where ambient aerosol was sampled through heated inlets using a high flow-rate impinger (Bertin Technologies, Model Coriolis $\mu$, France). The samples were analysed for the INPC immediately on site with a drop freezing technique using ETH's DRINCZ (Drop Freezing Ice Nuclei Counter Zurich - David et al. (2019)) in Wolfgang and University of Basel's LINDA (LED-based ice nucleation detection apparatus - Stopelli

et al. (2014)) at Weissfluhjoch as described in Mignani et al. (2020). The detection limit of the INPC (lowest concentration measurable) was calculated according to Vali (1971) considering the concentration obtained if only one (first) drop froze. A detection limit of $5.2 \cdot 10^{-4}\,\mathrm{L}^{-1}$ and $4.9 \cdot 10^{-4}\,\mathrm{L}^{-1}$ has been determined for Wolfgang and Weissfluhjoch, respectively.

## 2.2 HoloGondel

The HoloGondel platform consists of the HOLographic Imager for Microscopic Objects (HOLIMO 3G in Beck et al. (2017)),

which records the concentration, size distribution and shapes of cloud particles, a temperature and relative humidity sensor (HygroMet4, Rotronic) in a ventilated housing (RS24T, Rotronic) and a pressure sensor (Fig. 2). During the RACLETS campaign, the platform was installed on one of the gondolas running at the upper section between the Gotschnagrat mountain station (2280 m) and the middle station at Gotschnaboden (1790 m) covering a horizontal distance of about 830 m. To avoid any influence by the gondola stations on the measurements, data was only used if the difference of the pressure measured on

the gondola and the pressure measured at the stations was more than 1.5 hPa, corresponding to a vertical distance of more than 15 m and a total distance of about 30 m between the gondola and the stations. The gondola runs with a maximum speed of about $6\,\mathrm{m\,s}^{-1}$ leading to a total measurement time of about 140 s per ride. To avoid influences from the gondola and its swing arm, only measurements of uphill rides were analysed, when the setup was in front of the gondola in the direction of travel.

HOLIMO 3G is an open-path instrument, which uses digital in-line holography. Holograms of the sample volume between

the two towers (see Fig. 2b) are recorded of which $13.7\,\mathrm{cm}^3$ were considered for the analysis, producing a 3D distribution of the cloud particles between the two towers and a 2D image of each cloud particle. A more detailed description of the measurement principle can be found in Beck et al. (2017) and Henneberger et al. (2013). The pixel size is 3.1 μm, which allows us to observe cloud particles larger than 6.2 μm (Beck et al., 2017). The differentiation between ice and liquid is based on the shape of the particles (circular vs. non-circular). This is possible for particles larger than approximately 25 μm, depending on the shape of

the ice particles (Henneberger et al., 2013).

A neural network for particles larger than 25 μm (Touloupas et al., 2020) and a decision tree for particles smaller than 25 μm in their major axis size were used to separate cloud droplets from ice crystals and sort out artifacts, which are falsely identified as cloud particles by the software. All ice crystals were manually confirmed after the automated classification. Therefore, the uncertainty in the concentration of ice particles can be estimated with $\pm 5\,\%$ for ice crystals larger than about 100 μm and



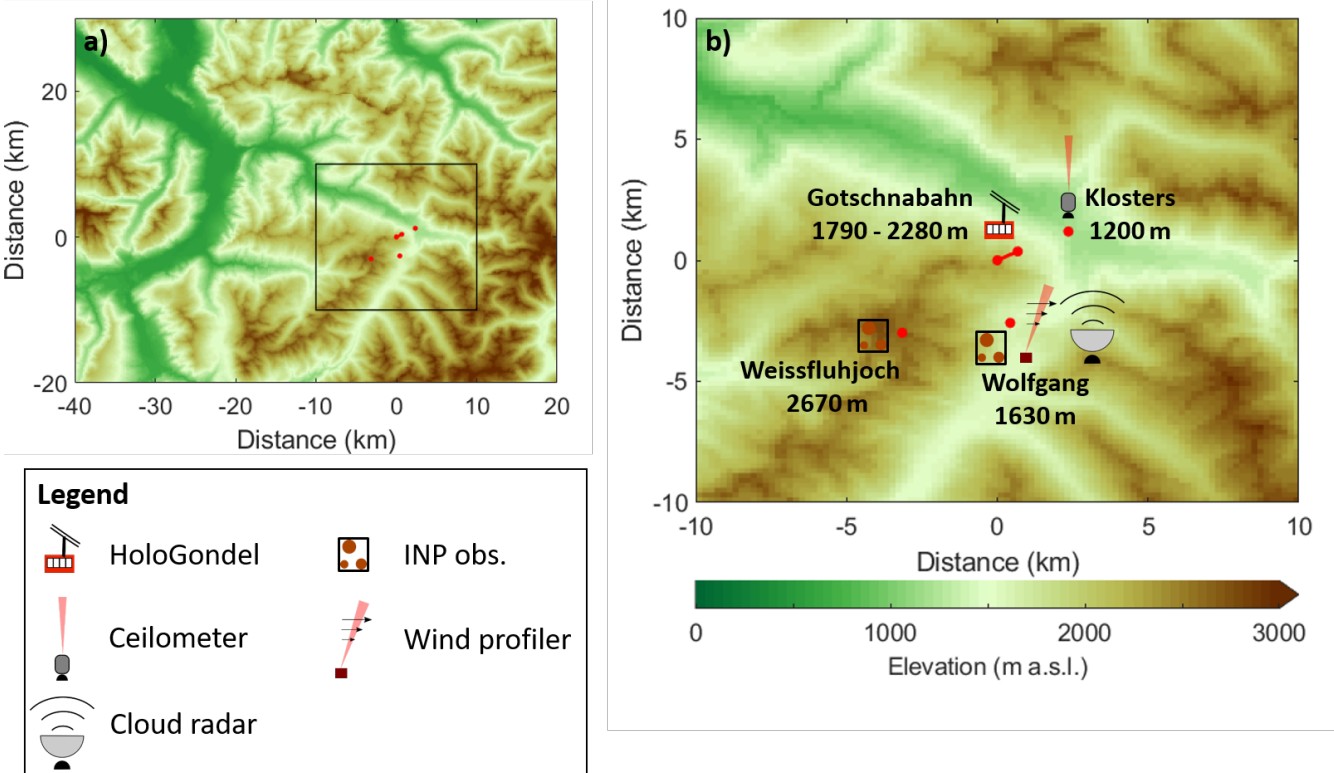

**Figure 1.** Overview of the measurement location and the experimental setup. The geographical location of the Gotschnabahn and the surrounding topography is shown in a). An enlarged section of the measurement sites (black rectangle in a)) and the instrument setup are shown in b). The elevation data was taken from the digital height model DHM25 of the Federal Office of Topography swisstopo: https://shop.swisstopo.admin.ch/de/products/height_models/dhm25200, last access: 9 March 2020.

$\pm\,15\,\%$ for ice crystals smaller than $100\,\mu m$ (Beck, 2017). All ice particles were also manually classified into the habits plates, columns, irregular, aged ice (rimed particles and aggregates) and unidentified referring to ice crystals, which could not be classified because they were too small or because their orientation did not allow a decision on their habit. For the uncertainty of the different habits, the counting uncertainty ($\sqrt{N}/V$; $N$: number of crystals, $V$: measurement volume) was added, because of
5  their relatively low number compared to the measurement volume. The uncertainty of cloud droplets is estimated to be $\pm\,6\,\%$ as determined for the classification with the neural network in Touloupas et al. (2020). Again, for droplets larger than $40\,\mu m$ the counting uncertainty was added due to their relatively small numbers.


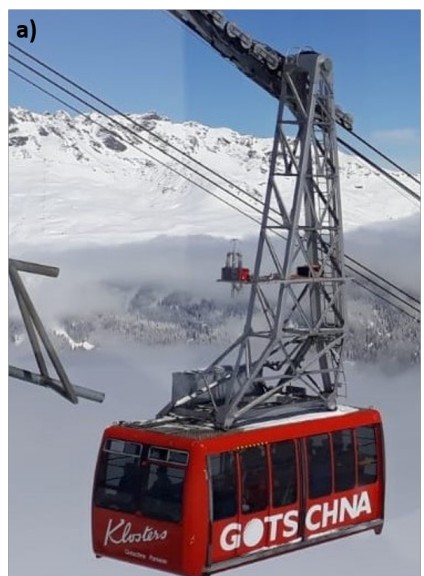
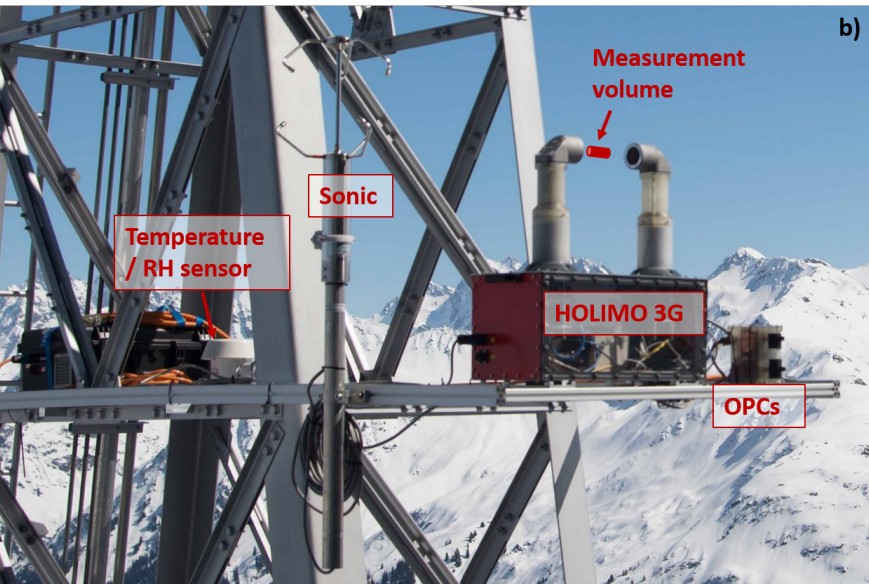

**Figure 2.** The HoloGondel platform installed on the Gotschnabahn in a) and an enlarged picture of the platform in b) showing HOLIMO 3G with its measurement volume between the two towers, two optical particle counters (not used in this study) next to HOLIMO and the temperature / RH sensors. The Sonic anemometer was turned off due to a limited supply of power.

## 3 Case study 22 February 2019

### 3.1 Weather situation

On 22 February 2019, Switzerland was located in front of a ridge, which guided air masses from the north to the Davos region. A trough over Russia reached relatively far to the south supporting the rise of air masses over Germany, Austria and eastern Switzerland, which were all covered by stratiform clouds. From the cloud radar measurements (Fig. 3), it can be anticipated that the cloud top height was about 4 km in the Davos region referring to a cloud top temperature of around -8 °C as measured by a radiosounde profile launched at 10:25 UTC from Wolfgang. Between 5 and 9 UTC, the cloud radar showed a midlevel cloud above the stratiform cloud, reaching from about 4.5 km to about 6.5 km (Fig. 3 a)) that might have acted as a seeder cloud in the early morning.

As can be inferred from the ceilometer data, light precipitation started at around 7 UTC in Klosters, where the valley station of the Gotschnabahn is located (Fig. 3 b). The cloud base was located slightly above the height of the Gotschnaboden during the precipitation period, which lasted until about 10:30 UTC when the cloud began to dissipate. Measurements with HoloGondel were taken between 8 and 10 UTC. The rides used for the analysis are shown as grey bars in Fig. 3. During the measurement period, the temperature in Klosters increased from about 1.5 °C to 3.5 °C and the temperature at Gotschnagrat stayed relatively constant slightly below -2 °C. The temperature at Gotschnaboden was derived from measurements on the gondola and is, therefore, only available when the gondola was in operation and close to Gotschnaboden. During the measurement period it





remained close to 0 °C. The melting layer can be inferred from the dark band of the ceilometer data (Sassen et al., 2005) below Gotschnaboden (Fig. 3 b)). A total of 9 complete measurement profiles were taken between 0 °C and -2.7 °C.

The main wind direction was from north to northeast as can be inferred from the wind profiler measurements taken in Wolfgang in Fig. 4 b). However, the ground measurements show that the wind direction was strongly influenced by the orography (Fig. 4 a)). The valley north of the Gotschnabahn forced the wind direction to northwest as can be seen from the wind measurements on Gotschnagrat. Thus, air masses measured on Gotschnabahn were pushed up the valley coming from northwest and must have been lifted through the melting layer before reaching the measurement site.

The slope inclination in the wind direction measured at Gotschnagrat is about 7° (averaged over Gotschnagrat and 1 km northwest). Assuming that the measured horizontal wind of about 5 m s$^{-1}$ (Fig. 4) was blown up the slope without friction, the vertical wind speed reached about 0.6 m s$^{-1}$. The eddy dissipation rate, which is a measure for turbulence, was calculated from the wind profiler and cloud radar measurements as described in Griesche et al. (2019) and reached values between 20 cm$^2$ s$^{-3}$ and 90 cm$^2$ s$^{-3}$ between 1800 m and 2300 m over Wolfgang between 8 and 10 UTC. Such values were reported in stratiform clouds (Borque et al., 2016) as well as in cumulus clouds with weak updrafts (Siebert et al., 2006). Since no measurements of turbulence are available over Klosters, we assume that similar turbulence is present at the measurement site.

## 3.2 In situ measurements

Between 8 UTC and 10 UTC, a total volume of 43.7 L was measured by HOLIMO, distributed over nine uphill rides, each contributing between 2.6 L and 8.2 L. The difference in the measurement volumes is due to different onsets and offsets of the automated recording. HOLIMO observed aged ice crystals, here defined as aggregates and rimed particles, irregular particles, plates and very few small columns (see Fig. 5). About 70% of of all ice crystals were classified as unidentified because they were too small or the particle orientation made a decision on its habit inconclusive. As discussed in section 4.1, unidentified particles as well as plates smaller than 93 µm are referred to as small plates. They are marked with dots in the histogram plot of the ice crystal size distribution in Fig. 5.

The concentrations of ice crystals and cloud droplets stayed relatively constant over the measurement period (see Fig. 6). Taking the whole measurement period into account, a mean cloud droplet number concentration (CDNC) of 156 cm$^{-3}$±9 cm$^{-3}$ and a mean ICNC of 6.0 L$^{-1}$±0.9 L$^{-1}$ was measured. Between the different rides the CDNC varied between 67 cm$^{-3}$ and 215 cm$^{-3}$ and the ICNC between 3.4 L$^{-1}$ and 18.0 L$^{-1}$. The most common observed ice particle habit was plates with a mean concentration of 0.8 L$^{-1}$±0.3 L$^{-1}$ varying between 0 L$^{-1}$ and 1.4 L$^{-1}$ between the different rides. Note that this is a lower estimate, since the software has difficulties in detecting transparent particles and depending on the size and orientation, plates may be hard to identify. Besides the ice particle habits, a mean concentration of droplets larger than 40 µm in diameter of 0.3 L$^{-1}$±0.1 L$^{-1}$ (see Fig. 7 for their size distribution), varying between 0 L$^{-1}$ and 0.8 L$^{-1}$ between the different rides, was observed. It should be mentioned that the missing observations of plates or large droplets during single rides could be a stochastic effect due to the relatively low concentrations compared to the measurement volumes of each ride, i.e. for the smallest measurement volume of 2.6 L$^{-1}$, one observed particle results in a particle concentration of 0.4 L$^{-1}$. In the following,





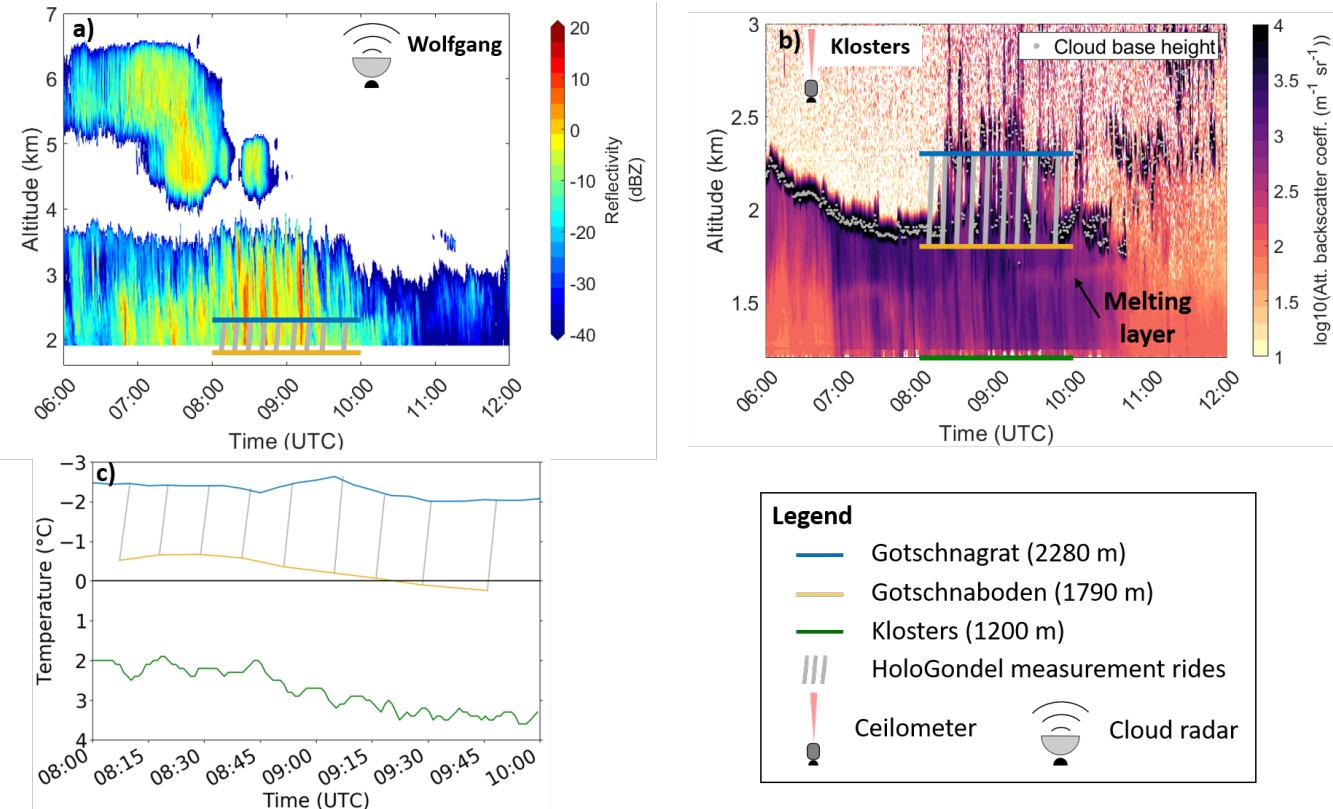

**Figure 3.** Observations of the reflectivity measured by the cloud radar installed in Wolfgang are shown in a). The ceilometer data in b) shows the estimated cloud base with grey dots above Klosters. A dark band underneath the cloud base indicates the melting layer. Temperature measurements at the Gotschnagrat, the Goschnaboden and Klosters, during the time HoloGondel was measuring, are shown in c).

we consider the whole measurement period and site as one measurement volume, where the cloud particle concentrations are constant.

The INPC was measured in the valley in Wolfgang (1630 m) and on the mountain top of Weissfluhjoch (2670 m) (see Fig. 1 for the geographical positions) simultaneously at 8:00 UTC and 10:00 UTC on 22 February 2019. No INP was detected for
5   temperatures higher than -6 °C at both sites. Therefore, the upper estimate of the INPC at the measurement site (T>-3 °C) is equal to the detection limit and on the order of $5 \cdot 10^{-4} \, \mathrm{L}^{-1}$ for both sites, which lies several orders below the measured ICNC.

## 4   Discussion

### 4.1   Estimation of the secondary ice crystal number concentration

Since the INPC at the measurement location lies several orders of magnitude below the ICNC, the contribution from primary
10  ice nucleation will be neglected. Therefore, the concentration of secondary ice crystals can be estimated from the concentration





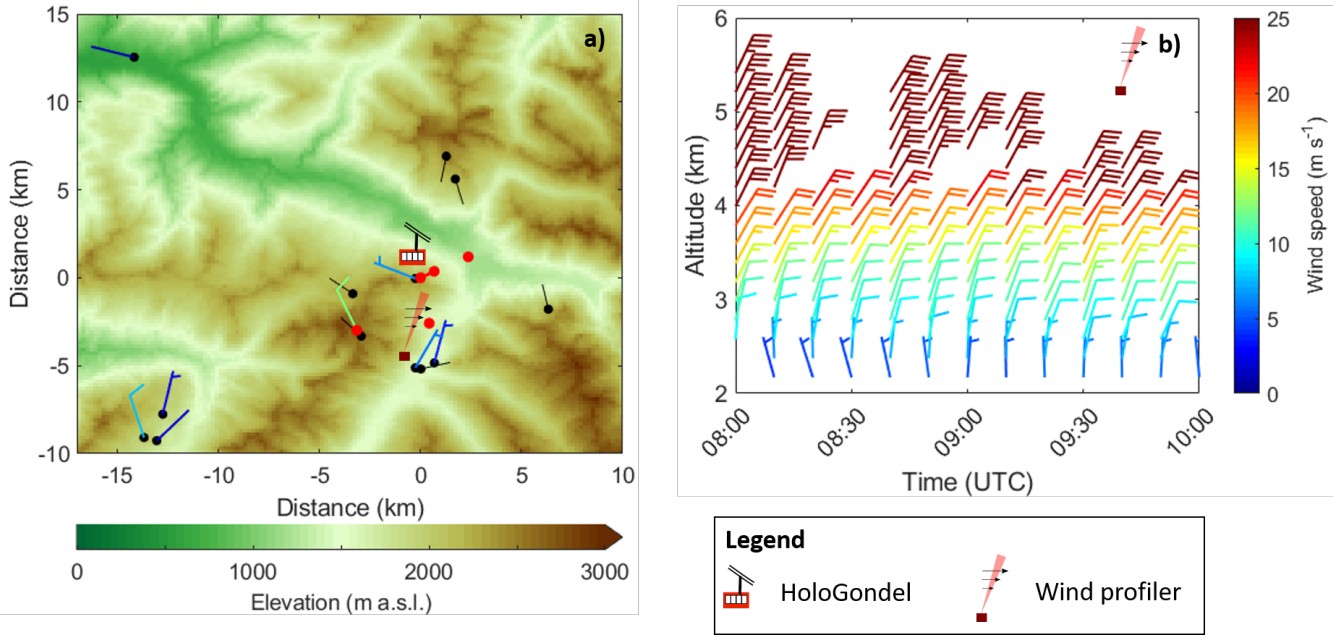

**Figure 4.** The wind direction and speed at different weather stations averaged between 8 and 10 UTC in the Davos region are shown in a). The colors of the wind barbs show the wind speed according to the colorbar, while stations with black lines only report wind direction. The wind profiler data, measured in Wolfgang is shown in b).

of ice crystals that have newly formed and grown at the same location. The measurements were taken at temperatures warmer than -3 °C. In this temperature regime, newly formed ice crystals grow into plates (Libbrecht, 2005; Bailey and Hallett, 2009). Figure 5 shows the shapes and the size distribution of the ice crystals divided logarithmically into 9 size bins. The histogram plot shows that the majority of the classified particles smaller than 93 μm were plates (54%; Fig. 5, unidentified particles excluded), while aged crystals can only be found above that size. Hence, we take 93 μm as a threshold to divide between newly formed ice crystals and those, which sedimented from above.

About 70% of the ice crystals smaller than 93 μm (from now on referred to as "small ice") could not be classified into a certain habit because the resolution was too low. Yet, a transparent part observed in the middle of many of these particles (see Fig. 5) supports the assumption based on the temperature regime that these ice crystals are actual plates. In the following, small ice classified as plates including the unidentified particles is referred to as "small plates".

A plate with a maximum dimension of 93 μm has a fall velocity of about $0.06 \, \mathrm{m \, s^{-1}}$ (using the equations given in Pruppacher (2010)) which is one order of magnitude lower than the estimated updraft of about $0.6 \, \mathrm{m \, s^{-1}}$. This supports the assumption that small plates are unlikely to have sedimented from above. About 4% of the small ice was classified as columns. The few observed columns have an aspect ratio close to 1 (see Fig. 5), which is typical for columns growing at temperatures just below -3 °C at low supersaturation (Libbrecht, 2005; Bailey and Hallett, 2009) and therefore, must have sedimented from slightly higher altitudes. This is possible because columns have a higher fall velocity than plates (a column with a length of 93 μm falls



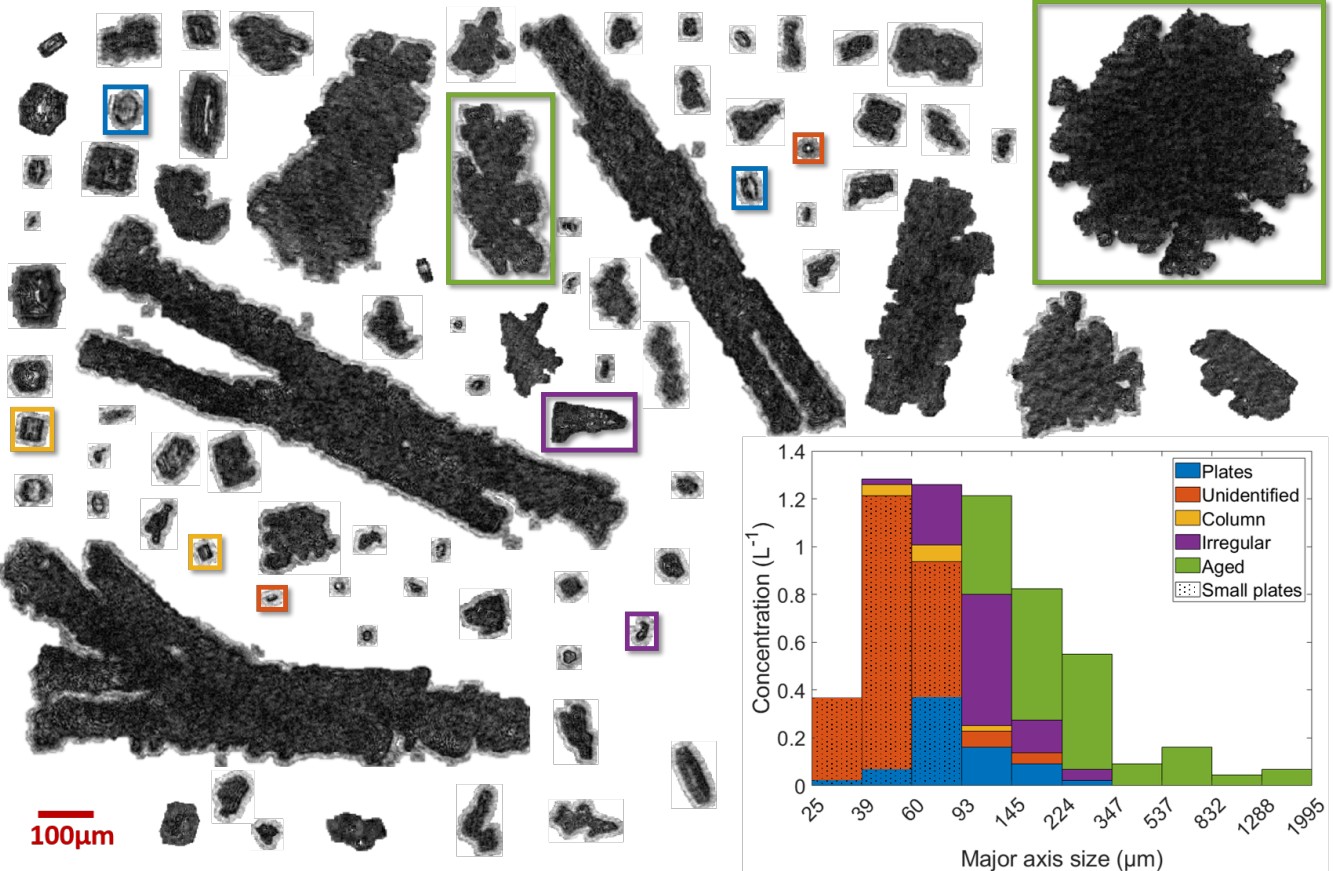

**Figure 5.** The Figure shows a randomly selected sample of ice crystals, which were observed in the cloud together with a stacked size distribution of the different habits. Unidentified crystals as well as plates smaller than 93 μm in their major axis size are referred to as "small plates" as discussed (see text for more details). A few examples are encircled with the color of the habit used in the size distribution plot.

with about $0.3\,\mathrm{m\,s^{-1}}$ using the equations given in Pruppacher (2010)). About 9% of the small ice was classified as irregular particles. These could be large secondary ice splinters. However, the fall velocity of irregular particles is hard to assess and it remains unclear if they have fallen from above or formed at the measurement site by SIP. Based on these assumptions, the lower estimate of the secondary ice concentration, which has formed and grown at the measurement site, is equal to the concentration

5     of small plates as defined above and is on the order of $2.6\,\mathrm{L^{-1}}\pm0.6\,\mathrm{L^{-1}}$.

The time a plate needs to grow to 93 μm is hard to assess at temperatures close to 0 °C because small variations in the environmental conditions, e.g., a temperature fluctuation of only 0.5 °C, can change the growth time on the order of several minutes. To calculate the growth time, we use the general equation of the ice particle growth by vapor diffusion given in Fukuta and Takahashi (1999). Using the mass-size relation of hexagonal plates from Mitchell et al. (1990) and assuming water vapor

10     saturation over liquid at a temperature of -2 °C, a splinter of 5 μm needs about 9 minutes to grow to a size of 93 μm and about





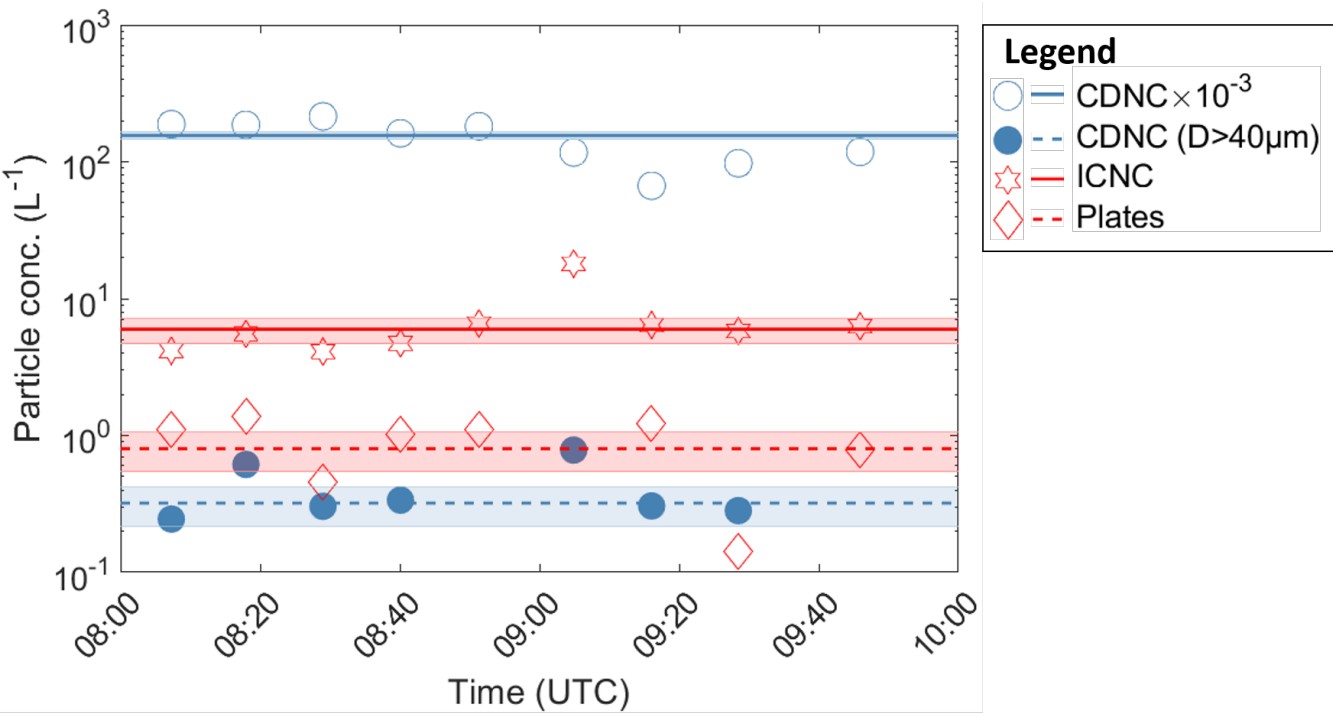

**Figure 6.** The ICNC, CDNC, CDNC for droplets larger than 40 µm and the concentration of plates averaged over each of the nine measurement rides are shown with the four different symbols. The corresponding lines show the mean concentration of the complete measurement volume with the shaded areas as the uncertainty of the mean concentrations.

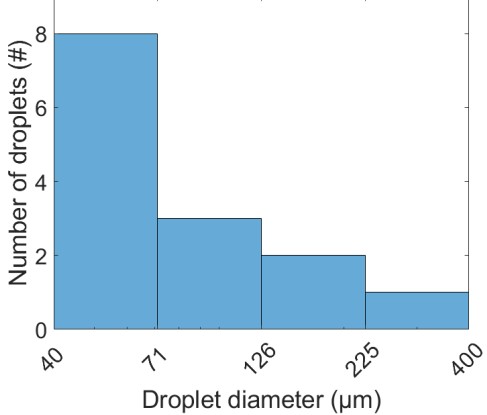

**Figure 7.** A histogram of the sizes of all observed droplets larger than 40 µm in diameter.





5 minutes to grow to a size of 60 μm. Thus, for any given time, a plate of 93 μm is 4 minutes older than a plate of 60 μm if the temperature is constant at -2 °C and the water vapor is saturated over water. Therefore, plates with sizes between 60 μm and 93 μm must have newly formed by SIP within a time span of 4 minutes. The same calculation can be done for a bigger size interval, e.g. plates between 39 μm and 93 μm, which must have newly formed by SIP during a time span of about 6 minutes.

Taking temperature variations in the measurement volume into account, i.e. varying the temperature between -2 °C and -1 °C, plates between 60 μm and 93 μm were newly formed by SIP during a time span of 4 to 6 minutes, while plates between 39 μm and 93 μm were newly formed by SIP during a time span of 6 to 10 minutes depending on the temperature. The observed concentration of plates with sizes between 60 μm and 93 μm was $0.9\,\mathrm{L}^{-1}\pm0.3\,\mathrm{L}^{-1}$, whereas between 39 μm and 93 μm the concentration of plates was $2.2\,\mathrm{L}^{-1}\pm0.5\,\mathrm{L}^{-1}$ averaged over the two-hour measurement period. Thus, $0.9\,\mathrm{L}^{-1}\pm0.3\,\mathrm{L}^{-1}$ of

secondary ice has formed within 4 to 6 minutes and $2.2\,\mathrm{L}^{-1}\pm0.5\,\mathrm{L}^{-1}$ within 6 to 10 minutes. Taking all named uncertainties into account, the rate of secondary ice production during our case study lies between $0.17\,\mathrm{L}^{-1}\,\mathrm{min}^{-1}$ and $0.3\,\mathrm{L}^{-1}\,\mathrm{min}^{-1}$.

### 4.2 Contribution of different SIP mechanisms

Droplets larger than 40 μm up to a size of 380 μm in diameter were observed between 0 °C and -2.7 °C (Fig. 7). Korolev et al. (2020) explained the occurrence of large droplets just above the melting layer with a recirculation process (see section 1.2).

Ice particles fall through the melting layer and melt into drizzle drops, which can be reintroduced into the cloud by sufficiently high updrafts. The estimated updraft in this case study is about $0.6\,\mathrm{m\,s}^{-1}$, which is equal to the fall speed of a 150 μm droplet (Rogers and Yau, 1989). Most of the observed droplets were smaller than 150 μm (Fig. 7), while the remaining ones could have brought into the cloud by local turbulences. Also note that the updraft velocity is a very rough estimate, increasing it by $0.2\,\mathrm{m\,s}^{-1}$ is already enough to lift 93% instead of 79% of the observed droplets.

Droplet fragmentation requires the presence of large droplets (>≈40 μm, (e.g., Lawson et al., 2015; Korolev et al., 2020)) and has no temperature constraint towards high temperatures. It is therefore a possible mechanism to explain the observed secondary ice. Collisional breakup cannot be completely ruled out but is not expected to produce high secondary ICNCs at warm temperatures as outlined in the introduction. Furthermore, the missing observation of fragment-like ice particles (e.g., broken-off branches) supports that collisional breakup was not very active. Moreover, fragments from collisional breakup are

not necessarily small particles and therefore, could be miscounted as aged or irregular ice in this study.

The rime-splintering process as well as ice fragmentation during thermal shock can both be excluded from being active because they require lower temperatures. MPCs are supersaturated with respect to ice, therefore, also the requirements for ice fragmentation during sublimation are not fulfilled (Korolev et al., 2020). Korolev et al. (2020) argued that INP activation in transient supersaturation around freezing drops could not be shown to be active in the atmosphere.

Therefore, we expect that most of the small plates emerged from droplet fragmentation and that the orographically-induced updraft serves as a constant supply of new droplets larger than 40 μm in diameter, which originate from melted ice crystals. A schematic of this recirculation process in mountainous regions is shown in Fig. 8. Because new droplets are continuously provided, we can assume that our measurements reflect a steady-state such that we observe freshly produced as well as aged





**Figure 8.** Schematic of the recirculation process, which was introduced by Korolev et al. (2020) and adapted for an updraft created by a mountain slope. Aged ice crystals sediment from the cloud and turn into drizzle sized drops when they fall through the melting layer. These drops are reintroduced into the cloud by wind forced up the mountain slope (updraft), where they freeze when they collide with aged ice crystals and fragment upon freezing. The small fragments grow into plates due to the environmental conditions (T>-3 °C) and may hit other droplets, which again freeze and possibly fragment.

particles simultaneously. The combination of SIP above the melting layer and high updrafts on the windward slope will transport the secondary ice crystals to higher altitudes where they influence the cloud microphyisics.

### 4.3 Parametrization of SIP by droplet fragmentation at warm temperatures

Here we derive a parametrization of the SIP by droplet fragmentation at temperatures close to 0 °C when primary ice nucleation can be neglected and droplets freeze only by the collision with ice crystals, which either sedimented from above or formed by SIP. Like Korolev et al. (2020), we assume that only droplets larger than 40 μm are likely to contribute to the SIP by droplet fragmentation. The average number of splinters generated per second per droplet larger than 40 μm ($G_{sp}$) is the product of the





droplet freezing rate by collision ($f_{\mathrm{col}}$), the droplet fragmentation probability during freezing ($p_{\mathrm{df}}$) and the number of splinters per fragmenting droplet ($N_{\mathrm{sp}}$):

$$G_{\mathrm{sp}} = f_{\mathrm{col}} \cdot p_{\mathrm{df}} \cdot N_{\mathrm{sp}} \tag{1}$$

The probability that a droplet with diameter $d$ and a fall velocity $v(d)$ collides with an ice crystal and freezes is equal to the collision efficiency $E$. It needs to be multiplied by the combined cross section of the ice crystal and the droplet and the relative velocity between those two to obtain the collection kernel. If we divide the ICNC in $i = 1, 2, ..., N$ size bins, we can approximate the probability that a droplet collides with any ice crystal by summing up the collection kernels of each size bin multiplied by the ICNC per size bin ($ICNC_i$). For simplification, the average fall velocity of the ice crystals in each size bin $\overline{v_i}$ and the average diameter of the ice crystals in the respective size bin $\overline{d_i}$ is used in the equation.

$$f_{\mathrm{col}}(d) \approx \sum_{i=1}^{N} E \cdot ICNC_i \cdot |\overline{v_i} - v(d)| \cdot \frac{\pi \cdot (d + \overline{d_i})^2}{2} \tag{2}$$

The probability of droplet fragmentation is size dependent (Takahashi and Yamashita, 1969; Kolomeychuk et al., 1975; Lauber et al., 2018). For a fragmentation to occur, the surface energy has to be overcome, which is proportional to $d^2$. Assuming a fragmentation probability of 40% for droplets with $d = 300\,\mu\mathrm{m}$ at temperatures larger than -2.5 °C (i.e. $p_{\mathrm{df}}(d{=}300\,\mu\mathrm{m}) = \mathrm{a} \cdot d^2 = 0.4$), as measured by Keinert et al. (2020), the droplet fragmentation probability per freezing droplet $p_{\mathrm{df}}$ can be estimated as

$$p_{\mathrm{df}}(d) \approx 4.4 \cdot 10^6 \cdot \mathrm{m}^{-2} \cdot d^2 \tag{3}$$

The concentration of splinters produced per fragmentation event could not be quantified until now because many of the splinters being produced during a fragmentation event might be too small to be observed with the available measurement techniques (Lauber et al., 2018; Keinert et al., 2020). Here we consider the maximum number of splinters observed during a breakup event for different droplet sizes, as given in Lauber et al. (2018) taken from different studies as the best estimate available. The data points suggest a linear correlation of the number of splinters being produced per droplet $N_{\mathrm{sp}}$ and the droplet diameter $d$. Applying a linear regression, $N_{\mathrm{sp}}$ can be estimated as:

$$N_{\mathrm{sp}}(d) \approx 9 \cdot 10^4 \cdot \mathrm{m}^{-1} \cdot d \tag{4}$$

The number of splinters produced per second by each droplet larger than 40 µm at temperatures close to 0 °C in the absence of INPs in MPCs is subsequently given by:

$$G_{\mathrm{sp}}(d) = f_{\mathrm{col}}(d) \cdot p_{\mathrm{df}}(d) \cdot N_{\mathrm{sp}}(d) \approx 6.2 \cdot 10^{11} \cdot \mathrm{m}^{-3} \cdot d^3 \cdot \sum_{i=1}^{N} E \cdot ICNC_i \cdot |\overline{v_i} - v(d)| \cdot (d + \overline{d_i})^2 \tag{5}$$

### 4.3.1 Application of the parametrization to the case study

In this section, the parametrization derived for SIP by droplet fragmentation for temperatures close to 0 °C in the absence of INPs is applied to the presented case study. For simplicity, we assume that the ICNC as well as the size and shape distribution of





the ice crystals stays constant over the whole measurement period of two hours, meaning that any ice crystal, which leaves the measurement volume is immediately replaced by a new one of the same size and shape. The same is valid for cloud droplets, which leave the measurement volume or freeze and potentially produce secondary ice splinters. Thus, we assume a constant production of secondary ice. Furthermore, we expect that droplets larger than 40 µm have a collision efficiency of $E = 1$.

Because of the broad size and shape distribution of the ice crystals, a rough assumption of their fall velocity for the estimation of the relative fall velocity between the droplets and ice crystals has to be made. Therefore, we divide the ice crystals in two size bins ($N = 2$) and assume that half of the ice crystals ($ICNC_1 = 3\,\mathrm{L}^{-1}$) are plates with a size of $\overline{d_1} = 50\,\mu\mathrm{m}$, while the other half ($ICNC_2 = 3\,\mathrm{L}^{-1}$) is lump graupel with a size of $\overline{d_2} = 300\,\mu\mathrm{m}$. Plates with $L \approx 50\,\mu\mathrm{m}$ fall with a terminal velocity of about $\overline{v_1} = 0.03\,\mathrm{m\,s}^{-1}$ (Pruppacher, 2010), while lump graupel with a diameter of 300 µm fall with about $\overline{v_2} = 0.7\,\mathrm{m\,s}^{-1}$

(Locatelli and Hobbs, 1974). The terminal velocity of the single droplets is calculated with the equations given in Rogers and Yau (1989) and are shown in Table 1 together with the calculated parameters for each droplet.

To calculate the production rate of secondary ice by droplet fragmentation, the average amount of splinters produced by all large droplets $\overline{G_{\mathrm{sp}}(d)}$ has to be multiplied by their concentration ($0.3\,\mathrm{L}^{-1}\pm0.1\,\mathrm{L}^{-1}$). Taking the single sizes of the droplets observed in this case study (Fig. 7) into account, this yields a production rate of $0.10\,\mathrm{L}^{-1}\mathrm{min}^{-1}\pm0.03\,\mathrm{L}^{-1}\,\mathrm{min}^{-1}$ of secondary

ice, which is below the estimated production rate of secondary ice of $0.17\,\mathrm{L}^{-1}\,\mathrm{min}^{-1}$ to $0.3\,\mathrm{L}^{-1}\,\mathrm{min}^{-1}$ derived from the observations.

The most uncertain parameter in eq. (5) is the number of splinters produced during a droplet fragmentation event. Taking the uncertainties into account and assuming that the proportionality of the number of splinters and the droplet diameter is correct, $N_{\mathrm{sp}}$ needs to lie in the range of $N_{\mathrm{sp}} \approx 1.2\cdot10^5\cdot\mathrm{m}^{-1}\cdot d$ to $N_{\mathrm{sp}} \approx 4.1\cdot10^5\cdot\mathrm{m}^{-1}\cdot d$ to produce the measured concentrations of

small secondary ice particles. This is equivalent to a range of 24 to 82 splinters produced by a fragmenting droplet of 200 µm in diameter and is up to 5 times higher than the first assumption of $N_{\mathrm{sp}}$.

Apart from the average amount of splinters produced, also the probability that a droplet fragments when it freezes might be different in the atmosphere from what has been measured in the laboratory. Keinert et al. (2020) showed that droplet fragmentation is significantly higher in moving air than in stagnant air. Therefore, it may be reasonable to assume that $p_{\mathrm{df}}$ is

even higher in turbulent conditions. Taking the uncertainties into account and assuming that all droplets larger than 40 µm will fragment upon freezing ($p_{\mathrm{df}}(d) = 1$), $N_{\mathrm{sp}}$ needs to lie between $N_{\mathrm{sp}} \approx 9\cdot10^4\cdot\mathrm{m}^{-1}\cdot d$ and $N_{\mathrm{sp}} \approx 2.2\cdot10^5\cdot\mathrm{m}^{-1}\cdot d$ to explain the measured secondary ice concentration. This is equivalent to a range of 18 to 43 splinters produced per fragmenting droplet of 200 µm in diameter. Thus, the first assessment of $N_{\mathrm{sp}}$ lies in the range of uncertainty if we expect that all droplets larger than 40 µm fragment when they freeze. Taking the average of the upper and lower estimate of $N_{\mathrm{sp}}$, the number of splinters

produced per droplet larger than 40 µm in diameter at temperatures close to 0 °C when turbulence or strong wind speeds are present would change to:

$$G_{\mathrm{sp}}(d) = f_{\mathrm{col}}(d) \cdot N_{\mathrm{sp}}(d) \approx 2\cdot10^5\cdot\mathrm{m}^{-1}\cdot d \cdot \sum_{i=1}^{N}\cdot ICNC_i \cdot |\overline{v_i} - v(d)| \cdot (d+\overline{d_i})^2 \tag{6}$$





**Table 1.** Contribution of the single droplets to the secondary ice concentration including the values of the different parameters used in eq. (5) and eq. (6) and the estimated fall velocity based on the equation given in Rogers and Yau (1989). Note that $p_{df} = 100\%$ for all droplets in eq. (6)

| $d$ (μm) | $v(d)$ (m s$^{-1}$) | $f_{col}$ (% min$^{-1}$) | $p_{df}$ (%) (eq. (5)) | $N_{sp}$ (#) (eq. (5), eq. (6)) | $G_{sp}$ (min$^{-1}$) (eq. (5), eq. (6)) |
|---|---|---|---|---|---|
| 42 | 0.05 | 2.2 | 0.8 | 3.8, 5.3 | $6 \cdot 10^{-4}$, 0.1 |
| 43 | 0.06 | 2.2 | 0.8 | 3.9, 5.4 | $7 \cdot 10^{-4}$, 0.1 |
| 43 | 0.06 | 2.2 | 0.8 | 3.9, 5.4 | $7 \cdot 10^{-4}$, 0.1 |
| 45 | 0.06 | 2.2 | 0.9 | 4.0, 5.6 | $8 \cdot 10^{-4}$, 0.1 |
| 47 | 0.07 | 2.2 | 1.0 | 4.2, 5.8 | $9 \cdot 10^{-4}$, 0.1 |
| 52 | 0.08 | 2.2 | 1.2 | 4.7, 6.5 | $1 \cdot 10^{-3}$, 0.1 |
| 53 | 0.08 | 2.2 | 1.2 | 4.7, 6.6 | $1 \cdot 10^{-3}$, 0.1 |
| 66 | 0.27 | 1.7 | 1.9 | 6.0, 8.3 | $2 \cdot 10^{-3}$, 0.1 |
| 71 | 0.29 | 1.7 | 2.2 | 6.4, 8.9 | $3 \cdot 10^{-3}$, 0.2 |
| 84 | 0.34 | 1.7 | 3.1 | 7.6, 10.5 | $4 \cdot 10^{-3}$, 0.2 |
| 115 | 0.46 | 1.5 | 5.8 | 10.4, 14.4 | $9 \cdot 10^{-3}$, 0.2 |
| 170 | 0.68 | 1.0 | 12.7 | 15.3, 21.2 | $2 \cdot 10^{-2}$, 0.2 |
| 202 | 0.81 | 2.2 | 18.0 | 18.2, 25.3 | $7 \cdot 10^{-2}$, 0.5 |
| 382 | 1.53 | 18.8 | 64.3 | 34.4, 47.8 | 4.2, 9.0 |

The calculated values for each parameter of eq. (5) and eq. (6) are shown in Table 1 for the single droplets. The contribution of the single droplets is highly unbalanced and the largest observed droplet (380 μm) alone is responsible for 97% and 79% of the produced secondary ice concentration using eq. (5) and eq. (6) respectively.

### 4.3.2 Caveats of the parametrization and its application to the case study

The first main caveat is the parametrization of $N_{sp}$. The proportionality of $N_{sp}$ to $d$ was based on only four data points, which were derived from three different studies using different measurement techniques and only show the maximum number of observed fragments as discussed in Lauber et al. (2018). Apart from this, there is no physical basis for this correlation. There are so far no reliable measurements of the average number of fragments being produced per fragmentation and recent work by Kleinheins et al. (2020) provides an indication that a majority of possible fragment ejections during freezing could not be

observed by the applied measurement techniques. The application to the case study suggests that the number of splinters may be up to 5 times higher than assumed in eq. (4) but critically depends on the largest observed droplet. Until further measurements can constrain the concentration of fragments produced per fragmentation, $N_{sp}$ remains highly uncertain.

    The second caveat is that the contribution of the different droplets is highly unbalanced. For example, the largest droplet ($\sim$380 μm) in the present case study contributes 97% to the total amount of the produced concentration of secondary ice when

using the generally derived parametrization (eq. (5)) and 79% when using the tuned parametrization (eq. (6)) (see Table 1). This



imbalance might be a real possibility and very few large droplets may be enough to explain high concentrations of secondary ice, while the contribution of droplets smaller than about 100 μm may be negligible. However, the observation of a single droplet with a certain size is statistically insignificant and more observations are needed to determine the real size distribution of the cloud droplets.

Thirdly, we assume a continuous flow over two hours without any vertical gradient for the application to the case study. Since the secondary ice splinters are expected to be very small, they will be lifted up with the updraft faster than the drizzle drops while they are growing to an observable size. The moment they leave the measurement volume before they reach a size of 93 μm, they will not be counted as secondary ice splinters anymore even though they were produced inside the measurement volume. Therefore, we expect that the secondary ice concentration is slightly underestimated. Drizzle drops can also produce

secondary ice when they are outside the measurement volume but the secondary ice particles can be lifted into the measurement. However, this is not very likely because measurements were taken very close to 0 °C and droplets are thus unlikely to freeze before they reach the measurement volume.

Lastly, the determination of the concentration of secondary ice was based on rather rough assumptions, e.g. small ice crystals, which could not be classified, were expected to be plates and the growth time was determined for specific environmental

conditions, which in reality changed with time and position inside the measurement volume. Apart from this, splinters were assumed to have a specific size, while they could vary in reality. Moreover, only splinters, which were small enough to grow into plates, were considered as secondary ice in this study.

## 5   Summary

On 22 February 2019, wind from northwest pushed air masses up a mountain slope where measurements were taken just above

the melting layer on a gondola (see Fig. 3). The measurements showed relatively constant conditions during the measurement period of two hours (Fig. 6) with a CDNC of about 160 cm$^{-3}$ and an ICNC of about 6 L$^{-1}$, which exceeded the measured INPC by several orders of magnitude. The majority of the observed small ice crystals (L<93 μm) were identified as plates (Fig. 5). As this is the preferred ice crystal habit at the temperatures in the measurement volume (T>-3 °C), ice crystals smaller than 93 μm are assumed to have newly formed at the same environmental conditions. At such warm temperatures, primary ice

nucleation can be neglected and the concentration of small plates (L<93 μm) most likely emerged from SIP.

Remarkable was the observation of relatively large droplets between 40 μm and 380 μm in diameter (Fig. 7) above the melting layer. The appearance of cloud droplets larger than about 40 μm is often connected to SIP by droplet fragmentation (Korolev et al., 2020). The rime-splintering process can be excluded to be active and only a small contribution by collisional breakup is assumed, leaving droplet fragmentation as the mainly responsible secondary ice process. A recirculation theory proposed

by Korolev et al. (2020) can explain these observations and can in general be applied to mountainous regions when a melting layer is present and sufficiently large updrafts are produced on the windward side by the local topography. Aged ice crystals fall through the melting layer as precipitation and melt into drizzle drops. If sufficiently large updrafts are present, these drops are blown up a mountain slope, lifted through the melting layer and refreeze if they collide with aged ice crystals. Due to the

pressure build-up during freezing, they will fragment and create secondary ice crystals, which again can initiate the freezing of another drizzle drop (see schematic in Fig. 8). The secondary ice crystals will be transported to higher altitudes where they influence the cloud microphyisics and subsequently the radiation budget, the lifetime and the precipitation pattern of the cloud.

A parametrization was introduced in section 4.3 for the generation of secondary ice particles by droplet fragmentation at tem-

peratures close to 0 °C when primary ice nucleation is basically absent (eq. (5)). Based on limited available measurements of former laboratory studies, it is assumed that the amount of splinters produced per droplet is linearly correlated with their diameter and that a droplet of 200 µm produces 18 splinters on average when it fragments. Applying the presented parametrization to our measurements could not explain the estimated concentration of secondary ice. However, if we assume that all droplets larger than 40 µm fragment when they freeze, the estimated generation of secondary ice lies in the range of uncertainty (i.e.; a

droplet of 200 µm in diameter produces between 18 and 43 splinters upon fragmentation). This assumption may be reasonable when strong wind speeds or turbulence are present. In either case, the application shows that droplets smaller than about 100 µm are negligible for SIP by droplet fragmentation. However, these results critically depend on a single large droplet next to other rather crude assumptions and should be considered with caution. Therefore, the parametrization should be implemented in a model and tested on other case studies, which observed high concentrations of small ice crystals in clouds with temperatures

close to 0 °C. The implementation can improve the prediction of the ICNC in mountainous regions where orographically forced updrafts through a melting layer are observed. A better prediction of the ICNC can improve the determination of the optical properties, the lifetime and the precipitation formation from clouds containing ice (e.g., Lohmann, 2002; Henneberg et al., 2017).

*Code and data availability.* Code and data is available on request.

*Author contributions.* AL analysed and interpreted the data, created the figures and wrote the manuscript with contributions from all authors. AL, JH, CM, FR, JP and JW performed the measurements on the Gotschnabahn. MH operated the radar wind profiler and the ceilometer and processed the data.

*Competing interests.* The authors declare that they have no conflict of interest.

*Acknowledgements.* The authors would like to thank the whole RACLETS team for the installation and maintenance of the measurement

setup as well as many fruitful scientific discussions. Especially we would like to thank Michael Lehning (WSL/SLF, EPFL) and his whole team for their effort in supporting and realizing the RACLETS campaign. A special thanks also goes to the Davos Klosters Bergbahnen AG and the staff of the Gotschnabahn, especially the technical managers Andrea Margadant and the managing director Markus Good for the permission to take measurements on one of their gondolas as well as the on-site support. We would also like thank Paul Fopp for providing his





land for the remote sensing measurements. We like to thank Alexander Beck for his support in the organization of the RACLETS campaign. We thank the Swiss Federal Office of Meteorology and Climatology MeteoSwiss for providing us with meteorological measurements and installing the ceilometer and the weather station in Klosters as well as the radar wind profiler in Wolfgang. We also like to thank the paragliding club Grischna for providing us with meteorological measurements from their Holfuy station at Gotschnagrat. We like to thank
5 Hannes Griesche for supplying us with turbulence data, Patric Seifert (TROPOS, Germany) for discussing their measurements, Benjamin Walter for providing as with data from the snow drift station at Gotschnagrat and we like to thank Pila Bossmann (wetterboss.com) for discussing the general weather situation. AL, FR, JW and UL acknowledge funding from the Swiss National Science Foundation (SNSF) grant number 200021_175824. CM acknowledges funding from the SNSF grant number 200021_169620.





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
