# Peer review of "Continuous secondary ice production initiated by updrafts through the melting layer in mountainous regions"

_Atmospheric Chemistry and Physics, 2020_

## Referee Comment (RC1) · Andrew Heymsfield (Referee) · 7 Nov 2020

Review of "Continuous secondary ice production initiated by updrafts through the melting layer in mountainous regions", by Lauber et al., acp-2020-986.

This is an interesting article that uses a gondola with a holographic camera (HOLIMO) to sample a cloud on a mountain slope just above the melting layer at temperatures warmer than -3 °C. Small hexagonal plate crystals were sampled, and given that the vertical motion was estimated to be about 0.6 m/s, this would imply that 1) secondary ice particles were produced in the temperature range 0 to -3C, and 2) that small drops (>40 microns) that shatter when freezing are thought to be responsible for the production of the ice crystals. Nine uphill sampling penetrations were made on 22 February 2019. Calculations based on laboratory measurements and ice crystal growth equations are used to support their conclusions.

Here are my primary concerns.

1) This is obvious but several cases that support your conclusions would have been desirable.

2) Many studies based on in-situ aircraft observations, especially those in tropical regions, have sampled updraft regions (some of them with weak vertical motions) with comparable size drops that have not identified secondary ice particles (SIP) from particle probes, including holographic imagers and the cloud particle imager (CPI) in this temperature range.

3) It seems entirely possible that snow that could fall through the updraft into the melting layer partially melted and created fragments that would have been carried up into the 0 to -3C temperature range and been incorrectly identified as SIP.

4) Given my comments below about ice crystal growth rates and terminal velocity, it seems unlikely that the droplets and SIP plates would have resided in this temperature range long enough to have grown to 60 microns and larger.

5) Although obvious, how applicable are the laboratory experiments of fragmentation applicable to natural clouds?

My more detailed comments appear below.

Page 10, 11. Is it possible that the small plates are a result of partially melted ice that fell through the melting layer and then partially melted ice fragments were carried up brought the melting layer by the 0.6 m/s updraft?

1, line 7: small plates at -3C? According to Fukuta and Takayashi (1999), the basic crystal habit is thick plates ($>-4.0°C$). I recommend providing that reference at this

point in the article.

1, line 11: high temperatures > slightly sub-0C temperatures.

2, 5: "water vapor pressure" to "relative humidity?

2, 12: "exist" to "can exist".

2, 13: I don't think its necessarily the primary ice that causes SIP.

11, line 6. You've calculated how long it takes to grow plates of up to 93 microns diameter at temperatures 0 to -3C. The linear growth rate is extremely slow, because the plates are "thick". The Mitchell et al. (1990) mass dimensional relationship for plates is therefore not applicable. What would the growth rate be if the ratio of the diameter to thickness is 1.0?. Please refer to Figure 10 of Fukuta and Takahashi, who give the appropriate axial dimensions. And their terminal velocity, which will govern how long they stay in the 0 to -3C temperature range before being lofted to higher altitudes and lower temperatures.

My comment above will obviously affect your calculated rate of SIP.

13, 20: The ice number concentration at temperatures below -12C or so are not too much higher than the IN concentration. Also, one does not see evidence from in-situ measurements that there are copious numbers of small plate-like ice crystals at temperatures below -12C that would suggest a vibrant SIP process.

15, 13: "larger" to "higher"

16, 5-6. Terminal velocity can be readily calculated for all ice crystal sizes, based on their shape from the holographic images.

16, 25: Is it even reasonable to assume that 40 micron droplets all freeze and produce splinters? There's no evidence for this from in-situ aircraft measurements.

Andy Heymsfield, NCAR

10.5194/acp-2020-986
Atmos. Chem. Phys. Discuss.
2020

Please also note the supplement to this comment:
https://acp.copernicus.org/preprints/acp-2020-986/acp-2020-986-RC1-
supplement.pdf

———————————————————————

---

## Referee Comment (RC2) · Alexei Korolev (Referee) · 12 Nov 2020

**Review of "Continuous secondary ice production initiated by updrafts through the melting layer in mountainous regions" by Lauber et al.**

**Overview**
The paper consists of two parts. The first part presents the results of in-situ observations of cloud particles from a HoloGondel platform, which allowed vertical travel (490m) along the mountain slope. The paper is based on the measurements of cloud particles shapes and their concentrations collected by HOLIMO 3G during the case study - when the melting layer was located between the highest and lowest points of the platform travel. The results from the measurements showed a significant number of pristine (monocrystalline) ice particles, predominantly plates, at subfreezing temperatures T>-3C. Based on the comparisons with the concentration of INPs it was concluded that the pristine ice crystals originate from secondary ice production (SIP). The main mechanism of SIP was assigned to the fragmentation of freezing large drops. The formation of large drops above the melting layer was explained by recirculation of melted ice through the freezing level.
In the second part, the authors explore the parameterizations of SIP based on the results of present observations and past lab studies.
I found the results of the first part undoubtedly interesting. If these results are to be confirmed by other research groups, then the role of the melting layer as a source of SIP should be reconsidered in cloud and NWPs simulations. However, the second part raised concerns, which are discussed in comments 8 and 9.
The paper deserves publication in ACP after addressing the comments listed below.

**Recommendation**: Accept after major revisions.

**Comments:**
1.  Visual assessment of the images in Fig.1 suggests that many pristine ice crystals (plates, thick plates, short columns, columns) were not identified as such and fall into a different category. This could occur due to their orientation (as mentioned in the text), which could hinder their classification. The eyeball recognition used in this study has a subjective component and it depends on the experience of the expert performing the recognition. A more objective way would be to use a neural network recognition trained on ice analogue crystals (e.g. Ulanowski et al. JQSRT, 2006) or synthetic images of pristine ice particles with different orientations. Developing this technique is obviously time consuming, and this is rather a suggestion for future research. Regarding this work, I am concerned that the number of pristine ice crystals were underestimated. Consequently, this may affect the parameterization, which you attempted in the second part of your paper. I would strongly suggest reassessing the number of pristine ice particles. For training purposes, you may consider a ray tracing software (e.g. Zemax or equivalent) to generate the appearance of facetted hexagonal ice crystals with different orientations.

2.  In addition to the previous comment, could you classify each particle in Fig.1. This will be useful for the assessment of the quality of image recognition and help understand the results of the particle classification.

3. Could you include your definition of an ice plate? What is the separation between thick plates and short columns in terms of their aspect ratios (h/L)?

4. Page 11: "*However, the fall velocity of irregular particles is hard to assess and it remains unclear if they have fallen from above or formed at the measurement site by SIP.*" You could use for the fall velocity assessment min-max range of the fall velocity based on the aspect ratio of ice particles and their sizes?

5. It would be useful to show the statistical significance of the amount of sampled cloud particles in a separate table, e.g. total number of sampled droplets, droplets >40um, total number of crystals, number of columns, plates, irregulars, aged ice, etc. The histogram in Fig.1 shows concentrations of ice particles with different habits. However, it is not clear what the statistical significance of these measurements.

6. It would be extremely beneficial and informative for this study to show the profiles of the air temperature, relative humidity, droplet concentration for each platform ascent, and show types of hydrometeors observed on different levels.

7. Page 13, line 16: "*The estimated updraft in this case study is about 0.6 m s$^{-1}$, which is equal to the fall speed of a 150 µm droplet*". 0.6m/s is a terminal fall velocity of 150µm diameter droplet. At the updraft velocity 0.6m/s this droplet will be suspended at the same altitude. In order to bring this droplet above the melting layer the updraft velocity should be $u_z$>0.6m/s.

8. Check Eq.2: E is a function of d and $d_i$. Summing should be performed over E as well. Not sure where 2 is coming from. Should it be 4?

9. The rate of splinter production is expected to depend on droplet concentration (CDNC). For example, if CDNC=0, then $G_{sp}$=0. However, none of the equations Eq.5 and Eq.6 includes CDCN. Please, check Eq.5 and Eq.6.

10. Page 8, line 19: "of of"

11. Page 13, line 28: "*Korolev et al. (2020) argued that INP activation in transient supersaturation around freezing drops could not be shown to be active in the atmosphere.*" This is an overstatement. The mentioned work suggested that this mechanism is unlikely to be active in a relatively warm environment (e.g. T>-4C). However, this mechanism may be active in convective clouds with strong updrafts at temperatures T<-20C.

12. Tale 1: remove duplicated line 3.

Alexei Korolev

---

## Author Comment (AC1) · 29 Jan 2021

We thank our reviewer Andy Heymsfield for this very thorough and constructive feedback. The incorporated suggestions significantly improved the quality of the manuscript. In the following, we address each comment and point to the according changes in our manuscript. The reviewer comments are displayed in italics, while the responses are given below each comment with the according changes in the manuscript in blue.

**General comments**

*1.      This is obvious but several cases that support your conclusions would have been desirable.*

We agree that several cases would have been desirable for a more conclusive analysis. However, the meteorological conditions (predominant wind direction in Klosters from southern/western direction) and instrumental issues did not allow for more observations under similar conditions during the RACLETS campaign. However, we plan to investigate this process in future campaigns and test the parameterization for other case studies.

*2.      Many studies based on in-situ aircraft observations, especially those in tropical regions, have sampled updraft regions (some of them with weak vertical motions) with comparable size drops that have not identified secondary ice particles (SIP) from particle probes, including holographic imagers and the cloud particle imager (CPI) in this temperature range.*

Thank you for this comment, which led to a more thorough literature research. However, we could not find any of the mentioned studies, which showed comparable droplet sizes but no SIP. On the contrary, all studies with similar conditions support our findings and a discussion of these studies was added to the manuscript (page 19, line 8-18):

**"4.4 Other case studies with similar observations**

Several other studies observed an increase in the concentration of small ice crystals at the presence of large supercooled drops in clouds (e.g., Stith et al., 2004; Lawson et al., 2015; Keppas et al., 2017; Korolev et al., 2020; Lloyd et al., 2020). Updrafts were made responsible for the origin of these large droplets in all of the studies. The measurements were taken mostly at temperatures lower than during our case study, where newly formed ice crystals grow into columns. The recirculation process through the melting layer described above (Fig. 7) is therefore also expected to play a role for droplet fragmentation in higher regions of the cloud if the updrafts are strong enough to lift drizzle drops high enough until they freeze. Besides droplet fragmentation, the rime-splintering process is expected to be active in the temperature regime between -3 °C to -8 °C, which makes the assignment of the observed secondary ICNC to a specific process more difficult. However, images of deformed frozen drops in all of the above mentioned studies strongly support that droplet fragmentation was active and should be accounted for besides the rime-splintering process."

*3.      It seems entirely possible that snow that could fall through the updraft into the melting layer partially melted and created fragments that would have been carried up into the 0 to -3C temperature range and been incorrectly identified as SIP.*

It is possible that ice crystals only partially melted and created fragments before they were carried up into our measurement regime, which is also a SIP process. We added the following paragraph to our manuscript to argue why we think this has a negligible effect (page 13, line 29 to page 14, line 2):

"Ice crystals can partially melt and create fragments, which can be lifted back into the cloud. However, the ice crystals have to be sufficiently small to be lifted, while at the same time, small ice crystals are likely to completely melt before re-entering the cloud. Furthermore, the breakup rates of this process depend on temperature and humidity and largely on the initial shape of the ice crystals (Korolev and Leisner, 2020). Oraltay and Hallett (1989) observed no sublimation breakup for columnar and plate-like crystals and breakup during melting only at relative humidities below 90%. The shapes of ice crystals in our study are mostly solid particles that have columnar and plate-like shapes (Fig. 5) and the relative humidity on the gondola never dropped below 95% (Fig. 3d). Taking all these considerations into account, we assume that ice fragmentation during melting has a negligible effect on the SIP."

*4.        Given my comments below about ice crystal growth rates and terminal velocity, it seems unlikely that the droplets and SIP plates would have resided in this temperature range long enough to have grown to 60 microns and larger.*

The responses are given to the following comments.

*5.        Although obvious, how applicable are the laboratory experiments of fragmentation applicable to natural clouds?*

The applicability of the laboratory experiments to natural clouds is very hard to assess at this point in time due to the lack of direct measurements inside natural clouds. The laboratory measurements are therefore only used as the best estimate available. We hope that the parameters can better be constrained with future measurements. They are easy to be adapted accordingly in our parametrization. We added the following paragraph to the manuscript to discuss this shortcoming (page 17, lines 20-23):

"The first main caveat is that the parametrization was derived solely from laboratory measurements. However, the direct observation of SIP by droplet fragmentation is basically impossible as the process happens on a millisecond timescale (Lauber et al.,2018) and the secondary ice splinters can be smaller than 10 µm (Korolev et al., 2020). Therefore, the laboratory measurements are the best estimate available."

**More detailed comments**

*1.      Page 10, 11. Is it possible that the small plates are a result of partially melted ice that fell through the melting layer and then partially melted ice fragments were carried up brought the melting layer by the 0.6 m/s updraft?*

See answer to general comment 3.

*2.      1, line 7: small plates at -3C? According to Fukuta and Takayashi (1999), the basic crystal habit is thick plates (>−4.0°C). I recommend providing that reference at this point in the article.*

According to Fukuta and Takahashi (1999), the basic ice crystal habits at temperatures above -4°C are indeed thick plates. However, they do not show any measurements above temperatures of -3°C. In our study, we observed thin plates at temperatures between 0°C and -2.7°C, which can be seen on the particle images of plates in Fig. 5 in the manuscript. Furthermore, Bailey und Hallett (2009) show a habit diagram of ice crystals including thin plates at temperatures warmer than -3°C. Therefore, we argue that the mass dimensional relationship for plates by Mitchell et. al. (1990) is applicable to our case.

*3.      1, line 11: high temperatures > slightly sub-0C temperatures.*

The text was adapted accordingly.

*4.      2, 5: "water vapor pressure" to "relative humidity?*

If the water vapor pressure is reduced, the relative humidity is also reduced at a given temperature. Therefore, the two terms are used interchangingly in this sentence. We prefer to describe it with water vapor pressure, as it is more consistent with the rest of the description.

*5.      2, 12: "exist" to "can exist".*

The text was adapted accordingly.

*6.      2, 13: I don't think it's necessarily the primary ice that causes SIP.*

We adapted the sentence to also account for secondary ice produced by secondary ice (page 2, line 13-15):

"The resulting so-called primary ice can create additional ice crystals (secondary ice), which again can fragment and produce more secondary ice by any kind of fragmentation referred to as secondary ice production (SIP) (e.g., Field et al., 2017)."

*7.     11, line 6. You've calculated how long it takes to grow plates of up to 93 microns diameter at temperatures 0 to -3C. The linear growth rate is extremely slow, because the plates are "thick". The Mitchell et al. (1990) mass dimensional relationship for plates is therefore not applicable. What would the growth rate be if the ratio of the diameter to thickness is 1.0?. Please refer to Figure 10 of Fukuta and Takahashi, who give the appropriate axial dimensions. And their terminal velocity, which will govern how long they stay in the 0 to -3C temperature range before being lofted to higher altitudes and lower temperatures.*

See answer to detailed comment 2.

*8.     13, 20: The ice number concentration at temperatures below -12C or so are not too much higher than the IN concentration. Also, one does not see evidence from in-situ measurements that there are copious numbers of small plate-like ice crystals at temperatures below -12C that would suggest a vibrant SIP process.*

Droplet fragmentation requires the presence of large droplets (>~40μm). In the absence of such large droplets, we do not expect droplet fragmentation to take place. In this study, we do not have measurements of cloud particles below temperatures of -12°C (our measurements were limited to a temperature range of 0°C to -2.5°C). Other measurements at such low temperatures with the presence of large droplets showed indeed a high concentration of small plates (e.g., Lawson et al. 2015, Korolev et al. 2020).

*9.     15, 13: "larger" to "higher"*

The text was adapted accordingly.

*10.     16, 5-6. Terminal velocity can be readily calculated for all ice crystal sizes, based on their shape from the holographic images.*

There are only equations available for specific ice crystal habits but not for irregular shaped ice crystals. To have a more accurate calculation of the relative velocities of ice crystals and cloud droplets, we now divide the crystals into plates and lump graupel and use given parametrizations to calculate the fall velocity of each ice crystal. This had a noticeable change only in the collision rate of the largest observed droplet (see Fig. 8 manuscript) by a factor of about 0.5. It had a minor effect on the splinter generation rate. See page 16, line 20-30 and all changes in the calculations hereafter:

"To calculate the fall velocities of the ice crystals, we divide them into plates and lump graupel. The former includes the classes plates and unidentified, while the later includes all other ice crystals (see section 2.2 for a more detailed description of the classes). The fall velocity of plates was calculated with the following equation from Pruppacher and Klett (2010) (converted to SI base units) using the maximum dimension of plates $L_{\mathrm{pla}}$:

$$v\left(L_{\mathrm{pla}}\right) \approx 156 \mathrm{m}^{0.14}\mathrm{s}^{-1} \cdot L_{\mathrm{pla}}^{0.86}. \tag{6}$$

To derive the fall speed of lump graupel with a maximum dimension of $L_{gra}$, we use the equation provided by Locatelli and Hobbs (1974) (again, converted to SI base units):

$$v\left(L_{gra}\right) \approx 124 \mathrm{m}^{0.34}\mathrm{s}^{-1} \cdot L_{gra}^{0.66}. \tag{7}$$

This yields a splinter generation rate of 0.06 L⁻¹ min⁻¹ ±0.02 L⁻¹ min⁻¹ of secondary ice, which is about one order of magnitude below the estimated production rate of secondary ice of 0.24 L⁻¹ min⁻¹ ±0.09 L⁻¹ min⁻¹ derived from the observations."

*11.      16, 25: Is it even reasonable to assume that 40 micron droplets all freeze and produce splinters? There's no evidence for this from in-situ aircraft measurements.*

The parametrization does not assume that all 40 μm droplets freeze and produce splinters. It provides a probability for both to happen. A 100 μm droplet for example will only freeze with a probability of 27% during the time it is being lifted through the complete measurement volume and when it freezes it fragments with a likelihood of 18% (see eq. (4) in the manuscript). (A 100 μm droplet has a fall velocity of about 0.4ms⁻¹ and will be lifted with 0.2 ms⁻¹ when an updraft of 0.6 ms⁻¹ is present. To be lifted up 490 m it will thus take 2450 s and the collision rate of a 100 μm droplet is $f_{col}$(100 μm) = 1.1e-4s⁻¹, see eq. (3) in the manuscript).

**Additional remarks**

We would like to point out that we did some essential changes, which are not all part of the responses. They are addressed in the following:

1. We removed Figure 7, which showed a histogram of the sizes of all observed droplets larger than 40 µm as all droplets are now shown in Fig. 8 of the manuscript, where the values of the different parameters are plotted.

2. The observed secondary ice production rate is given as a number with uncertainties instead of a range as this is easier to interpret (page 13, lines 8-9):

   "Taking all named uncertainties into account, the rate of secondary ice production during our case study is 0.24 $L^{-1}$ $min^{-1}$±0.09 $L^{-1}$ $min^{-1}$."

   This change slightly influenced all calculations, which included the observed secondary ice production rate in section 4.3.1.

3. The total splinter generation rate is given per volume and time instead of only per time as this makes its interpretation easier. Furthermore, we no longer consider size bins for the ice crystals but take all ice crystals into account to have more accurate calculations. This changed the equations in section 4.3.

4. The collision rate given in eq. (3) was off by a factor of 0.5 in the old version of the manuscript. This change led to different results and slightly different interpretations. Changes can be found in the abstract, section 4.3.1 and the summary. The following lines of the manuscript were adapted:

   Page 1, lines 12-15:

   "Based on previous measurements, we estimate that a droplet of 200 µm in diameter produces 18 secondary ice crystals when it fragments upon freezing. The application of the parametrization to our measurements suggests that the actual number of splinters produced by a fragmenting droplet may be up to an order of magnitude higher."

   Page 20, lines 10-13:

   "Applying the presented parametrization to our measurements could not explain the estimated concentration of secondary ice and the number of splinters produced per fragmenting droplet has to be higher, i.e., a droplet of 200 µm in diameter has to produce 99±62 splinters upon fragmentation. This number can be reduced to 44±26 if we assume that all droplets larger than 40 µm fragment when they freeze."

   All changes are marked in the final version of the manuscript.

**References**

Bailey, Matthew P., and John Hallett. 2009. "A Comprehensive Habit Diagram for Atmospheric Ice Crystals: Confirmation from the Laboratory, AIRS II, and Other Field Studies." *Journal of the Atmospheric Sciences* 66 (9): 2888-2899. doi:10.1175/2009jas2883.1.

Fukuta, Norihiko, and Tsuneya Takahashi. 1999. "The Growth of Atmospheric Ice Crystals: A Summary of Findings in Vertical Supercooled Cloud Tunnel Studies." *Journal of the Atmospheric Sciences* 56 (12): 1963-1979. doi:10.1175/1520-0469(1999)056<1963:Tgoaic>2.0.Co;2.

Korolev, A., I. Heckman, M. Wolde, A. S. Ackerman, A. M. Fridlind, L. A. Ladino, R. P. Lawson, J. Milbrandt, and E. Williams. 2020. "A new look at the environmental conditions favorable to secondary ice production." *Atmos. Chem. Phys.* 20 (3): 1391-1429. doi:10.5194/acp-20-1391-2020.

Lawson, R. P., S. Woods, and H. Morrison. 2015. "The Microphysics of Ice and Precipitation Development in Tropical Cumulus Clouds." *Journal of the Atmospheric Sciences* 72 (6): 2429-2445. doi:10.1175/JAS-D-14-0274.1.

Mitchell, David L., Renyi Zhang, and Richard L. Pitter. 1990. "Mass-Dimensional Relationships for Ice Particles and the Influence of Riming on Snowfall Rates." *Journal of Applied Meteorology and Climatology* 29 (2): 153-163. doi:10.1175/1520-0450(1990)029<0153:Mdrfip>2.0.Co;2.

---

## Author Comment (AC2) · 29 Jan 2021

We thank our reviewer Alexei Korolev for this very thorough and constructive feedback. The incorporated suggestions significantly improved the quality of the manuscript. In the following, we address each comment and point to the according changes in our manuscript. The reviewer comments are displayed in italics, while the responses are given below each comment with the according changes in the manuscript in blue.

**Comments**

*1.        Visual assessment of the images in Fig.1 suggests that many pristine ice crystals (plates, thick plates, short columns, columns) were not identified as such and fall into a different category. This could occur due to their orientation (as mentioned in the text), which could hinder their classification. The eyeball recognition used in this study has a subjective component and it depends on the experience of the expert performing the recognition. A more objective way would be to use a neural network recognition trained on ice analogue crystals (e.g. Ulanowski et al. JQSRT, 2006) or synthetic images of pristine ice particles with different orientations. Developing this technique is obviously time consuming, and this is rather a suggestion for future research. Regarding this work, I am concerned that the number of pristine ice crystals were underestimated. Consequently, this may affect the parameterization, which you attempted in the second part of your paper. I would strongly suggest reassessing the number of pristine ice particles. For training purposes, you may consider a ray tracing software (e.g. Zemax or equivalent) to generate the appearance of facetted hexagonal ice crystals with different orientations.*

We are aware that the eyeball recognition has a subjective component. A neural network recognition would be for sure more objective. However, it would need a large enough training set from our instrument, which is not available. Artificially created ice crystals or data from other instruments are most likely too different from the holographic images to be used for training. We are currently working on collecting enough data from holographic imagers to train a neural network. However, this will not be done within the time scope of the manuscript. Therefore, the eyeball recognition is so far our best estimate.

All ice crystals, which could not be clearly identified by their shape (i.e. "unidentified" class in Fig.5 in the manuscript), were expected to be plates as described in the manuscript. Therefore, we do not believe that the number of pristine ice crystals was underestimated. The parametrization was derived independently of our observation but based on laboratory studies and is, therefore, not influenced by our estimation of the number of pristine ice crystals. In a second step, we tuned the parametrization to explain our observations. The tuned version of the parametrization is undoubtedly influenced by the subjective component of the eyeball recognition. However, the shortcomings of this application are discussed in section 4.3.2.

*2.        In addition to the previous comment, could you classify each particle in Fig.1. This will be useful for the assessment of the quality of image recognition and help understand the results of the particle classification.*

We agree that showing the class of each displayed ice crystal is useful for the assessment of the quality of the image recognition. We adapted Fig. 5 in the manuscript to show the class of each displayed ice crystal.

*3.    Could you include your definition of an ice plate? What is the separation between thick plates and short columns in terms of their aspect ratios (h/L)?*

We added a definition of plates and columns (page 7, lines 4-6):

"Plates and columns are hexagonal prisms with the diameter of the hexagonal basis $a$ and the prism height $h$. Plates have a dimension of $a > h$, while columns have a dimension of $a \leq h$.

*4.    Page 11: "However, the fall velocity of irregular particles is hard to assess and it remains unclear if they have fallen from above or formed at the measurement site by SIP." You could use for the fall velocity assessment min-max range of the fall velocity based on the aspect ratio of ice particles and their sizes?*

There are only equations available for specific ice crystal habits but not for irregular shaped ice crystals. To have a more accurate calculation of the relative velocities of ice crystals and cloud droplets, we now divide the crystals into plates and lump graupel and use given parametrizations to calculate the fall velocity of each ice crystal. This had a noticeable change only in the collision rate of the largest observed droplet (see Fig. 8 manuscript) by a factor of about 0.5. It had a minor effect on the splinter generation rate. See page 16, line 20-30 and all changes in the calculations hereafter:

 "To calculate the fall velocities of the ice crystals, we divide them into plates and lump graupel. The former includes the classes plates and unidentified, while the later includes all other ice crystals (see section 2.2 for a more detailed description of the classes). The fall velocity of plates was calculated with the following equation from Pruppacher and Klett (2010) (converted to SI base units) using the maximum dimension of plates $L_{pla}$:

$$v(L_{pla}) \approx 156 \mathrm{m}^{0.14} \mathrm{s}^{-1} \cdot L_{pla}^{0.86}. \tag{6}$$

To derive the fall speed of lump graupel with a maximum dimension of $L_{gra}$, we use the equation provided by Locatelli and Hobbs (1974) (again, converted to SI base units):

$$v(L_{gra}) \approx 124 \mathrm{m}^{0.34} \mathrm{s}^{-1} \cdot L_{gra}^{0.66}. \tag{7}$$

This yields a splinter generation rate of 0.06 L$^{-1}$ min$^{-1}$ ±0.02 L$^{-1}$ min$^{-1}$ of secondary ice, which is about one order of magnitude below the estimated production rate of secondary ice of 0.24 L$^{-1}$ min$^{-1}$ ±0.09 L$^{-1}$ min$^{-1}$ derived from the observations."

*5.    It would be useful to show the statistical significance of the amount of sampled cloud particles in a separate table, e.g. total number of sampled droplets, droplets >40um, total number of crystals, number of columns, plates, irregulars, aged ice, etc. The histogram in Fig.1 shows concentrations of ice particles with different habits. However, it is not clear what the statistical significance of these measurements.*

We added the total concentration of all habits and their uncertainties to the histogram plot in Fig. 5 in the manuscript (page 11). The uncertainty for droplets larger than 40 μm and the CDNC is shown in Fig. 6 as shaded areas.

*6.       It would be extremely beneficial and informative for this study to show the profiles of the air temperature, relative humidity, droplet concentration for each platform ascent, and show types of hydrometeors observed on different levels.*

We agree that profiles over time would be beneficial. The profiles of the temperature and relative humidity can now be inferred from Fig. 3c and d in the manuscript and are described in the text as follows (page 8, lines 7-12):

"The temperatures and relative humidities at Gotschnaboden and Gotschnagrat (Fig. 3c, d) are derived from measurements of the highest and lowest point of the measurements on the gondola and are, therefore, only available when the gondola was in operation and close to one of the stations. During the measurement period, the temperature in Klosters increased from about +1.5 °C to +3.5 °C at a relative humidity of about 80%, while the measurements on the gondola were taken mainly between 0°C and -2.5°C at relative humidities above 95%."

We cannot analyse the ice crystal data profiles due to a lack of statistical significance but can show the CDNC for each ride divided into three height intervals (see Fig. 1 below) as cloud droplets are abundant enough for statistical significance. The figure shows that the lowest height interval has on average a lower CDNC, most likely because it is close to or partly at the cloud base. Furthermore, the habits number concentrations (excluding columns due to their total small amount) and the CDNC divided into three temperature intervals are shown averaged over all rides in Fig. 2 below. We divided the data in temperature intervals instead of height intervals to ensure the comparison of similar thermodynamical conditions. The figure shows that aged ice crystals are rather constant over the measured temperature range, while the smaller ice crystals are less abundant at higher temperatures in accordance with our theory: New ice crystals need some time to grow to a size where they can be detected and are being lifted up during this time. No secondary ice is expected to form above 0°C and therefore, we expect less secondary ice closer to 0°C as is already outlined in the manuscript (page 18, lines 6-10). At the same time, aged ice crystals fall from above and are not influenced by this effect. Even though, we agree that these plots are interesting to look at, we do not think that they provide essential information for our study and we therefore did not include them in our manuscript.

[Figure]

*Figure 1: The CDNC of each ride averaged over three height intervals.*

[Figure]

*Figure 2: The different habits, the ICNC and the CDNC divided into three temperature intervals containing the data from all rides.*

*7.     Page 13, line 16: "The estimated updraft in this case study is about 0.6 m s⁻¹, which is equal to the fall speed of a 150 μm droplet". 0.6m/s is a terminal fall velocity of 150μm diameter droplet. At the updraft velocity 0.6m/s this droplet will be suspended at the same altitude. In order to bring this droplet above the melting layer the updraft velocity should be uz>0.6m/s.*

We are not saying that the updraft is enough to lift a droplet of 150 μm in diameter but only droplets smaller than 150 μm in diameter. Most of the droplets are smaller than 150 μm (79%).  We also want to point out, that the given updraft velocity is a very rough estimate. We adapted the following sentence to make this more clear (page 13, lines 15-16):

"Most of the observed droplets were smaller than 150 μm and could be lifted up in the clouds by the updraft, while the remaining ones could have been brought into the cloud by local turbulences."

*8.     Check Eq.2: E is a function of d and di. Summing should be performed over E as well. Not sure where 2 is coming from. Should it be 4?*

Yes, E is a function of the droplet and ice crystal size and we added the dependency in the equation. Above that, we are very thankful for indicating a major error in our equation and replaced 2 by 4. This correction changed the results and the interpretation slightly. The following lines of the manuscript were adapted:

Page 1, lines 12-15:

"Based on previous measurements, we estimate that a droplet of 200 μm in diameter produces 18 secondary ice crystals when it fragments upon freezing. The application of the parametrization to our measurements suggests that the actual number of splinters produced by a fragmenting droplet may be up to an order of magnitude higher."

Page 20, lines 10-13:

"Applying the presented parametrization to our measurements could not explain the estimated concentration of secondary ice and the number of splinters produced per fragmenting droplet has to be higher, i.e., a droplet of 200 μm in diameter has to produce 99±62 splinters upon fragmentation. This number can be reduced to 44±26 if we assume that all droplets larger than 40 μm fragment when they freeze."

9.    *The rate of splinter production is expected to depend on droplet concentration (CDNC). For example, if CDNC=0, then Gsp=0. However, none of the equations Eq.5 and Eq.6 includes CDCN. Please, check Eq.5 and Eq.6.*

Eq. 5) and 6) gave the rate of splinter production per droplet with diameter *d*. To calculate the actual splinter generation rate, the equations needed to be multiplied with the concentration of droplets with according diameters, which was done to calculate the splinter generation rate in the cloud. However, we agree that presenting the equation in this way is misleading. To show the actual splinter generation in the cloud, the splinter generation per droplet with diameter $d_n$ is now summed up and divided by the total volume (page 15, lines 1-11 and all changes in the calculations hereafter):

"Here we derive a parametrization of the SIP by droplet fragmentation at temperatures close to 0 °C when primary ice nucleationcan be neglected and droplets freeze only by the collision with ice crystals, which either sedimented from above or formed by SIP. Like Korolev et al. (2020), we assume that only droplets larger than 40 μm are likely to contribute to SIP by droplet fragmentation. To calculate the splinter generation rate of a droplet with diameter $d_n$ ($g_{sp}(d_n)$), the droplet freezing rate by collision ($f_{col}(d_n)$) has to be multiplied by the droplet fragmentation probability during freezing ($p_{df}(d_n)$) and the number ofbsplinters per fragmenting droplet ($N_{sp}(d_n)$):

$$g_{sp}(d_n) = f_{col}(d_n) \cdot p_{df}(d_n) \cdot N_{sp}(d_n). \tag{1}$$

To obtain the total splinter generation rate ($G_{sp}$) in a volume $V$, which contains $N_{d>40\mu m}$ droplets with diameters larger than 40 μm, the sum of $g_{sp}(d_n)$ over all droplets with diameter $d_n$ ($n$= 1,2,..., $N_{d>40\mu m}$) has to be divided by $V$:

$$G_{sp} = \frac{1}{V}\sum_{j=1}^{N_{d>40\mu m}} g_{sp}(d_n). \tag{2}$$

10.    *Page 8, line 19: "of of"*

The double wording has been removed.

11.    *Page 13, line 28: "Korolev et al. (2020) argued that INP activation in transient supersaturation around freezing drops could not be shown to be active in the atmosphere." This is an overstatement. The mentioned work suggested that this mechanism is unlikely to be active in a relatively warm environment (e.g. T>-4C). However, this mechanism may be active in convective clouds with strong updrafts at temperatures T<-20C.*

The statement was adapted as follows (page 13, lines 26-27):

"Additionally, INP activation in transient supersaturation around freezing drops is unlikely to be active in a relatively warm environment (Korolev et al., 2020)."

12.     *Tale 1: remove duplicated line 3.*

The table showed the properties of each measured droplet of which two of them have the same diameter and were therefore displayed twice. However, we noticed that this is not a good way to display the results. We removed the table and included plots to get a better overview of the results (page 18: Fig. 8).

**Additional remarks**

We would like to point out that we did some essential changes, which are not all part of the responses. They are addressed in the following:

1.  We removed Figure 7, which showed a histogram of the sizes of all observed droplets larger than 40 µm as all droplets are now shown in Fig. 8 of the manuscript, where the values of the different parameters are plotted.

2.  The observed secondary ice production rate is given as a number with uncertainties instead of a range as this is easier to interpret (page 14, lines 2-3):

    "Taking all named uncertainties into account, the rate of secondary ice production during our case study is 0.24 $L^{-1}$ $min^{-1}$±0.09 $L^{-1}$ $min^{-1}$."

    This change slightly influenced all calculations, which included the observed secondary ice production rate in section 4.3.1.

3.  The splinter generation rate is given per volume and time instead of only per time as this makes its interpretation easier. Furthermore, we no longer consider size bins for the ice crystals but take all ice crystals into account to have more accurate calculations. This changed the equations in section 4.3.

    All changes are marked in the final version of the manuscript.